# Safety Evaluation of Nanotechnology Products

**DOI:** 10.3390/pharmaceutics13101615

**Published:** 2021-10-04

**Authors:** Abraham J. Domb, Ghorbanali Sharifzadeh, Victoria Nahum, Hossein Hosseinkhani

**Affiliations:** 1The Centers for Nanoscience and Nanotechnology, Alex Grass Center for Drug Design and Synthesis and Cannabinoids Research, School of Pharmacy, Faculty of Medicine, Institute of Drug Research, The Hebrew University of Jerusalem, Jerusalem 91120, Israel; victoria.nahum@mail.huji.ac.il; 2Department of Polymer Engineering, School of Chemical Engineering, Universiti Teknologi Malaysia, Johor Bahru 81310, Malaysia; hatef.sharifzadeh@gmail.com; 3Innovation Center for Advanced Technology, Matrix, Inc., New York, NY 10029, USA

**Keywords:** nanomaterials, nanoparticles toxicity, nanomedicine, toxicity assessment, oxidative stress, necrosis, apoptosis, biodistribution, cell viability, in vivo fate

## Abstract

Nanomaterials are now being used in a wide variety of biomedical applications. Medical and health-related issues, however, have raised major concerns, in view of the potential risks of these materials against tissue, cells, and/or organs and these are still poorly understood. These particles are able to interact with the body in countless ways, and they can cause unexpected and hazardous toxicities, especially at cellular levels. Therefore, undertaking in vitro and in vivo experiments is vital to establish their toxicity with natural tissues. In this review, we discuss the underlying mechanisms of nanotoxicity and provide an overview on in vitro characterizations and cytotoxicity assays, as well as in vivo studies that emphasize blood circulation and the in vivo fate of nanomaterials. Our focus is on understanding the role that the physicochemical properties of nanomaterials play in determining their toxicity.

## 1. Introduction

During the last 20 years, a significant growth in nanomaterial research has occurred, based on their numerous applications in medicine, photonics, and electronics [1,2,3,4,5]. Nanomaterials have been developed and used in a wide variety of commercial products, for example, electronic sensors, energy equipment, sun creams, and biomedical devices.

Toxicity research is important in order to fully understand the basic interaction of nanomaterials with natural tissues, because nanotechnology has a significant impact in the consumer and biomedical realms. One must understand the exact role of nanomaterials in vivo. When nanoparticles (NPs) are administrated into the human body, the effect of NP-loaded therapeutics, followed by their distribution into tissues (e.g., the kidneys, lung, and liver) must be precise. The clearance of nanomaterials can potentially occur through the pulmonary, hepatic, and renal systems. Thus, predicting long-term behavior becomes particularly difficult. NPs may aggregate in various organs and interact with off-target cells. They may degrade and be eliminated from the body due to their very small size [6]. There are, however, only a few studies examining whether nanomaterials are bio-compatible with natural organs, tissues, or cells. This review discusses the chemical, physical, and biological properties of various natural and synthetic nanomaterials, focusing on recent technologies related to the fabrication and processing of nanomaterials in different systems, emphasizing possible toxicity in in vitro and in vivo assays. We discuss the in vivo role of nanomaterials, including routes of administration, biodistribution, metabolism, routes of clearance, as well as blood biocompatability. The impacts of the physicochemical properties of nanomaterials and their toxicity are highlighted.

## 2. Mechanism of Toxicity

Nanoparticles can induce toxicity both in vitro and in vivo through various mechanisms. Examples of the most common mechanisms are oxidative stress, cell death mechanisms (apoptosis, autophagy, and necrosis), genotoxicity, and immunological responses. Each mechanism can lead to NP toxicity in a different pattern. The different mechanisms of NP toxicity are explained below.

### 2.1. Oxidative Stress

Oxidative stress is an imbalance between the production of reactive oxygen species (ROS) and antioxidant mechanisms [7] that can be explained as an increase in the generation of ROS or a decrease in antioxidants. A significant antioxidant found in animals, plants, and fungi is glutathione (GSH), which can halt losses caused by ROS. It is available in oxidized (GSSG) and reduced (GSH) forms. In usual healthy cases, most of the glutathione pool (about 90% of the total) is in a reduced state. An increased GSSG-to-GSH ratio indicates oxidative stress [8]. ROS can be triggered through various mechanisms, for example: (i) interaction with oxidative organelles such as mitochondria, (ii) involvement with redox active proteins, (iii) the chemical reaction of surface groups or coating from the NPs in the acidic environment, and (iv) the activation of different signaling routes via interaction with cell surface receptors [7,9]. In contrast, such reactions do not occur in the presence of all NPs, such as cerium oxide (CeO_2_). CeO_2_ NPs do not cause the formation of ROS, and they display a preserving effect against ROS damage both in vitro and in vivo [10,11]. Research shows that vitamin-C-conjugated NPs protect cells from oxidative stress at micromolar concentrations, and they induce cell death at millimolar concentrations [12]. The exact molecular target and chemical properties of oxidative stress, as well as the question of how it affects the modification of distinct biological procedures in cells subjected to engineered NPs (ENPs), have not yet been fully understood. Several studies postulate that reversible oxidative posttranslational modifications of protein cysteines via ROS and RNS show the basic mechanism of cell signaling that adjusts the protein functions of several cellular activities [13,14]. Protein S-glutathionylation (SSG) is a vital redox adjustment, altering apoptosis, mitochondrial metabolism, and transcription [15]. These SSG alterations are controlled by physiological properties. They can be repaired by glutaredoxin (Grx) enzymes [15,16]. Site-specific profiling of SSG modifications at a proteome-wide scale can be achieved through quantitative redox proteomics [17]. Using an SSG mechanism, Duan et al. assessed the modification of macrophage innate immune functions by ENPs via quantitative redox proteomics for the site-specific measurement of SSG alterations [18]. Three ENPs (silicon oxide (SiO_x_), Fe_3_O_4_, and cobalt oxide (II) (CoO)) were applied to stimulate disrupted macrophage function and cellular ROS, yielding low, moderate, and high propensity, respectively. SSG regulations indicate specific Cys residues and a wide range of redox-sensitive proteins corresponding to the total amount of cellular oxidative stress (CoO > Fe_3_O_4_ >> SiO_2_). ENPs that generate moderate and serious ROS show different pathways in response to SSG. Pathways modify protein stability and translation representative of the ER stress response. Proteins in phagocytosis, however, are highly susceptible to SSG in the presence of Fe_3_O_4_ [18]. ENPs trigger a subcytoxic degree of redox stress. ENP mitochondrial energetic pathways and classical stress responses are affected by SSG alternation moreso than redox stress caused by CoO.

### 2.2. Cell Death Mechanisms

This process is caused by cell replacement, or takes place when old cells are dying. Normally, it begins with organelle stress [19]. NPs can induce death-associated signaling via their interaction with various membranes or proteins in organelles. Specific proteins (receptors) involved in cell death are triggered by various NPs [20]. Cell death can be classified into three major mechanisms: apoptosis, autophagy, and necrosis.

#### 2.2.1. Apoptosis

Apoptosis, alternatively known as type-1 cell death, is a process of programmed cell death that is critical for the development and preservation of tissue homeostasis [21]. Wang et al. demonstrated the inherent mechanism of caspase-dependent apoptosis with both intrinsic and extrinsic pathways on bone tumor impressions via selenium [22]. In vivo data analysis showed that selenium-doped hydroxyapatite NPs (Se-HNPs) cause tumor apoptosis to suppress tumor growth and reduce toxicity. Membrane integrity is inhibited during apoptosis (contrary to necrosis), since many molecules that are considered “eat me” signals in the plasma membrane of dying cells exist [23]. Pardo et al. clarified the molecular mechanism of apoptosis in mice for laser PTT [24]. Apoptosis induced via the intrinsic/mitochondrial pathway was regulated through the Bak and Bax pathways by activating the BH3-only protein Bid using laser-irradiated hot gold nanoprisms (NPRs). The NPR-mediated apoptosis obtained was related to caspase-3 and caspase-9 proteins. Another molecular mechanism was presented by Hou et al. for in vivo photodynamic therapy via the mitochondrial apoptosis pathway [25]. They applied an NIR light base on TiO_2_-modified upconversion NP (UCNP) core/shell nanocomposites (UCNPs@TiO_2_ NCs). NaYF_4_:Yb^3+^,Tm^3+^@NaGdF_4_:Yb^3+^ core/shell UCNPs effectively converted NIR light to UV emissions. The UCNPs@TiO_2_ NCs generate intracellular ROS in the presence of NIR irradiation, thus reducing the mitochondrial membrane potential to release cytochrome c in the cytosol, leading to the activation of caspase 3, and causing the apoptosis of the cancer cell. Instead of UV light, NIR-based UCNPs@TiO_2_ exhibits deeper tissue penetration and inhibits tumor growth more effectively [25].

#### 2.2.2. Autophagy

Lin et al. have recently reported that tetrahedral DNA nanostructures (TDNs) containing a concentration of 250 nM improve autophagy and upregulate various autophagy-related proteins and genes [26]. TDNs are internalized through chondrocytes in the absence of other auxiliary agents. They focus on the cytoplasm of chondrocytes; however, only few single-strand DNAs (ssDNAs) are internalized by cells. TDNs increase autophagy via the PI3K/AKT/mTOR (mammalian target of rapamycin) pathway.

Several studies have revealed that NPs can disturb autophagy, which may lead to severe toxicological problems [27]. Additional research is required to establish the exact role of NPs in the induction of autophagy. This induction by a broad range of NPs (e.g., polymer [28], nanogel [29], and iron oxide NPs) has been studied [30]. Most of the examples cited are in vitro studies based on NP-induced autophagy that fail to imitate the complex in vivo physiological conditions. Zhou et al. evaluated QD-induced autophagy in living organismsm employing *Caenorhabditis elegans* (*C. elegans*) as the model organism [31]. They triggered autophagy in intestinal cells through the addition of QDs into the intestinal cells of *C. elegans* via oral feeding or microinjection. Figure 1 illustrates the real-time in situ tracking of autophagosome formation in live organisms.

This in vivo QD-induced autophagy is a defensive strategy of the organism aiming to degrade and recycle damaged cellular organelles. Fan et al. developed an autophagy regulation strategy based on folic-acid-modified N-doped carbon dots (FA-CDs) for visualized tumor therapy [32]. Stable FN-CDs exist in autophagic vacuoles in tumor cells. Their specific cellular uptake rate is a maximum of 93.40%. Cellular metabolism is influenced by FN-CDs that cause autophagy. Autophagic vacuoles are opened through autophagy inhibitors, and they release NP-activated signaling pathways, thereby killing tumor cells. This strategy shows therapeutic efficacy in 26 types of tumor cell lines. In vivo results indicate a greater efficiency (98%) than in traditional treatment (68%). Real-time in situ image tracking of the treatment has been reported through the use of FA-CDS.

#### 2.2.3. Necrosis

Liu et al. reported that graphene oxide (GO) triggered necrosis in macrophages through the activation of Toll-like receptor 4 (TLR4), thus signaling autocrine TNF-α production [33]. The prevention of TLR4 signaling with a selective inhibitor almost completely inhibits cell death. GO-induced necrosis of macrophages is partially related to the RIPK1-RIPK3 complex necrosis associated with TNF-α induction. GO treatment leads to massive intracellular ROS generation. GO uptake into macrophages causes a reduction in the ability to perform phagocytosis and the disruption of the cytoskeleton. Zhang et al. reported that single-walled carbon-nanohorns (SNH) and conventional carbon nanotubes (CNT) induce cell death via necrosis and apoptosis [34]. They showed that CNT triggers 1.9–4.8-fold increases in necrosis compared to SNH, thereby indicating that SNH initiates significantly lower necrosis inducibility (Figure 2). Therefore, SNH and CNT trigger cell death through pyroptosis-mediated necrosis (pyroptosis is a commonly regulated necrosis pathway), and SNH always induces lower cell death than CNT via apoptosis or necrosis.

Among programmed cell death processes, necrosis is considered the most common cellular response during tumor ablation performed through NP-mediated PTT [35]. In general, uncontrolled necrosis causes the release of intracellular constituents into the extracellular milieu. Therefore, induced responses may generate a harmful knock-on impact by damaging adjacent tissue, triggering tumor growth, etc. [36]. Loss of membrane integrity occurs particularly in necrotic cells. However, when considering in vitro environments, apoptotic cells also suffer from membrane integrity loss (secondary necrosis) in the absence of phagocytic cells. This behavior occurs later than during primary necrosis.

### 2.3. Genotoxicity

The destruction of genetic material in a cell due to toxicity is usually referred to as genotoxicity [37]. This includes DNA or chromosomal damage, including gene mutations, chromosome rearrangements, and breaks [38] using sophisticated signal transduction pathways [39]. Nanomaterials may influence genotoxicity because of their unique properties, for example, their very small size and broad surface area. Several factors are involved in the interaction of genetic materials with CNTs, e.g., similarity in size to that of microtubules, the capability of CNTs to penetrate the nuclear membrane, and the high affinity of SWCNTs to the G-C-rich areas in DNA sequences [40]. Reportedly, MWCNTs and SWCNTs triggered DNA damage in vitro in mesothelial cells [41]. NPs generate genotoxicity directly (primary genotoxicity) or indirectly (secondary genotoxicity). Inflammatory responses and oxidative stress cause secondary genotoxicity; primary genotoxicity may occur through an interaction of NPs with DNA or the mitotic apparatus [42]. Indirect genotoxicity of NPs can occur via the production of ROS through inflammatory cells. The resultant ROS can interact with DNA, ending in DNA oxidation, breakage, and/or lipid-peroxidation-mediated DNA adducts [43].

Genotoxicity can be detected in vitro by the Ames test, comet assay, and the chromosome aberration test, and can also be detected in vivo (rodent carcinogenicity, chromosome aberration, and mutation of endogenous genes). These assays are generally based on the detection of induced damages. Numerous NPs are reportedly genotoxic, such as polymeric and engineered NPs [44,45]. Many human cells have been used to evaluate the genotoxicity of CNTs. These include fibroblasts and RAW264.7 macrophages, normal lung epithelial BEAS-2B and SAEC cells, normal and malignant mesothelial cells, as well as malignant alveolar epithelial A549 cells. They show that high doses (>50 to 100 μg mL^−1^) of MWCNT and SWCNT induce genotoxicity. CNTs disrupt their dynamics and functions by imitating microtubules [46]. The Ames test indicates a sensitivity lower than that of micronucleus assays in terms of the genotoxicity assessment of AgNPs in human lymphoblastoid (TK6) cells [47]. One can attribute lesser sensitivity of the Ames test, which employs bacterial cells to evaluate mutagenicity, to non-endocytic bacterial effects. For this reason, the Ames test is not suitable for specific NPs [48]. The micronucleus assay can also yield false results because of reagent interference (i.e., cytochalasin B) with the cellular uptake of NPs [49]. DNA damage caused by NPs has also been measured by means of 8-oxoguanine (8-oxo-dG) ELISA [50]. This assay attains high throughput with a 96-well plate design; however, it measures only a biomarker of DNA oxidation, 8-oxo-dG, and does not reveal other DNA damage such as alkali-sensitive sites, double- and single-strand breaks, and abasic sites. The comet assay is the most common test for assessing the genotoxicity of nanomaterials. It has low reproducibility and throughput because of sample-to-sample variation (one sample displayed different results when tested on two separate glass slides) [51]. Watson et al. introduced a high-throughput screening (HTS) assay based on the CometChip approach that provides assessments of abasic sites, alkali-sensitive sites, and single-stranded DNA breaks in TK6 and adherent Chinese hamster ovary (H9T3) cells [52]. The researchers noted dose-dependent increases in cytotoxicity and DNA damage through the exposure to engineered nanoparticles (ENPs) (Ag, CeO_2_, Fe_2_O_3_, SiO_2_, and ZnO) at a dose range of 5, 10, and 20 μg/mL. The genotoxicity of ENPs in TK6 cells at 4 h and in H9T3 cells at 24 h was ZnO > Ag > Fe_2_O_3_ > CeO_2_ > SiO_2_, and Ag > Fe_2_O_3_ > ZnO > CeO_2_ > SiO_2_, respectively (Figure 3).

### 2.4. Immune Response

A wide range of NPs can stimulate, suppress, and/or elicit no immune response. Interaction with immune responses is associated with physiochemical properties that include shape, size, and charge [53]. NPs are able to adsorb, encapsulate, and/or chemically immobilize antigens within their surfaces, thus resulting in controlled release and delivery [54,55]. Niikura et al. demonstrated that various shapes of AuNPs modified with Nile virus envelope (E) protein cause cytokine secretion behavior in dendritic cells. Cubic and spherical AuNPs lead to the secretion of pro-inflammatory cytokines at high levels, but rod AuNPs cause the secretion of inflammasome-related cytokines interleukin (IL) 1β and IL-18 [56]. Long-aspect-ratio ENPs, (e.g., CNTs and nanowires) induce the activation of the inflammasome (secondary to oxidative stress) and induce a shape-dependent effect at the lysosomal level [57]. NPs may also activate cell and humoral immune responses through interaction with cell receptors and plasma proteins. They can also be internalized via mammalian cells. NPs can deliver sufficient antigens to targeted sites and enhance immune responses via their favorable bioavailability and cellular uptake, thereby reducing adverse effects [58].

Nanomaterials can be designed as vaccines to improve immune responses by reducing the unwanted activation or suppression of immune systems. This is suitable for infectious diseases and the immunotherapy of tumors [59]. Nanovaccines can regulate their hydrophilicity, shape, antigen loading, surface potential, and scale through various fabrication procedures and materials [59]. They promote long-term immune responses via adhesion, mixing, conjugation, and encapsulation of the antigen with NPs for vaccine delivery [60]. Pulendran et al. developed synthetic polymer NPs loaded with hemagglutinin (HA) from Toll-like receptor ligands and avian influenza H5N1. This is effective against swine influenza virus strains and lethal avian influenza in mice. It also causes robust immunity against H1N1 influenza [61]. Despite these NP-based vaccines, there are examples indicating that non-vaccine nanomaterials can also induce immune response [62]. Zhai et al. designed crosslinked (X-NPs) and non-crosslinked self-assembled NPs (NX-NPs) for antimetastasis and antiproliferation of melanoma [63]. Chondroitin sulfate (CS)-adipic dihydrazide (ADH)-chlorin e6 (Ce6)-lipoic acid (LA) loaded docetaxel (DTX) works as a redox/enzyme/ultrasound. Response conjugates form self-assembled NPs in the presence of aqueous solutions that are suitable for combining sonodynamic therapy (SDT) and chemotherapy. X-NPs induce the generation of ROS, leading to mitochondrial damage, followed by cell apoptosis via the mitochondria-caspase pathway. CD44 nanoscale distribution and targeted properties initiate the successful tumor homing ability of NPs, thereby causing potential antimetastasis and antitumor efficiencies, with declined expression of tumorous uPA and COX-2 proteins. The SDT of X-NPs triggers an immunological response via cell death. It causes the release of tumor-associated antigens.

## 3. Toxicity Assessment of Nanomaterials

### 3.1. In Vitro Characterization

The in vitro evaluation of biocompatibility with blood components is a necessary part of early preclinical development. Many research projects have reported NPs’ hemolytic properties. It is probable that the biological activity of NPs depends on physicochemical parameters that are not routinely considered in toxicity screening studies. Physicochemical properties may be important in comprehending the toxic effects of test materials. These properties include particle size and size distribution, shape, crystal structure, chemical composition, agglomeration state, surface area, surface chemistry, and charge, as well as porosity. In vitro techniques allow specific biological and mechanistic pathways to be isolated and tested under controlled conditions that are not feasible in in vivo tests. Various tests have been suggested for portal-of-entry toxicity for lungs, skin, and mucosal membranes and to target organ toxicity for the endothelium, liver, nervous system, heart, blood, spleen, and kidney. Non-cellular assessment of NPs’ durability, protein interactions, complement activation, and pro-oxidant activity have also been considered. Section 3.1 highlights the most basic in vitro characterizations.

#### 3.1.1. Electron Microscopy

Horvath et al. examined the in vitro cytotoxicity of boron nitride nanotubes (BNNTs) using SEM [64]. Nanotubes attached to the surface of exposed cells can cause toxicity by activating signaling pathways. SEM, therefore, has been applied to study the distribution of BNNTs on the surface of macrophages (RAW 264.7) and epithelial cells (A549). BNNT-treated A549 cells reveal no significant morphological changes, whereas many nanotubes bind to the cell surface in BNNT-treated macrophages. SEM indicates the ultrastructural modification of the A549 epithelial cell surface in the response of ENMs. The biocompatibility and dynamical processes of the interaction of nanomaterials with macrophage cells dosed with metal oxide NPs (CeO_2_, TiO_2_ and ZnO) were examined in another study through field emission scanning electron microscopy (FE-SEM) [65]. FE-SEM confirmed the interaction and the exact details of the internalization, adsorption, and ultrastructural location of NPs via the main constituents of the innate immune system. FE-SEM offers new insights into the biocompatibility and biodistribution of NPs with immune cells. SEM was also used to determine tumor margins based on the actual visualization of anti-AuNR-epidermal growth factor receptors (EGFRs) [66]. The distribution of EGFRs in and around tumor cells was evaluated using air SEM (airSEM), a highly sensitive form of microscopy for biological samples. AirSEM provides localized and actual visualization of the AuNR on tissues. The distance between healthy and tumor regions is within a radius of ~1 mm (Figure 4a).

Transmission electron microscopy (TEM) utilizes electrons transmitted through a sample to form images. TEM reveals considerable information for in vitro NP localization and uptake through the visualization of NPs’ locations in a cell/tissue, together with other techniques [67]. Both SEM and TEM are highly suitable for the characterization of nanomaterials before and after administration within samples. Several drawbacks limit their application: long sample preparation time with a low throughput, limited use of biological samples with thickness of 50–100 nm, and structural damage to the sample during preparation. Even so, TEM has been applied as a method in the cellular localization of various NPs [67,68]. It is also capable of detecting morphological changes on the cells caused by NPs. Ghandehari and co-workers utilized TEM for the cellular internalization of silica nanoparticles (SNPs) composed of nonporous spherical (Stöber) SNPs with diameters of 46 ± 4.9 nm (Stöber50), 432 ± 18.7 nm (Stöber500), and mesoporous spherical (MSN) particles with a diameter of 466 ± 86 nm (MSN500) [69]. The SNPs were localized inside vesicles situated in the cytosol of macrophages. TEM imaging shows that the number of Stöber500 particles attached to the outer part of cell was higher than that of MSN500 particles. Compared to Stöber500 SNPs, the higher internalization of MSN500 is sub-cytotoxic and causes enlargement of macrophage cells. Both nonporous and mesoporous spherical SNPs were applied to study the gene expression of cells. Stöber SNPs have a minor effect on gene expression. MSN, however, can change gene transcription without exhibiting any acute toxicity (Figure 4b). Scanning transmission electron microscopy (STEM) is a modified type of TEM that forms images by utilizing the combination of suitable detectors. Valiyaveettil et al. used STEM to investigate the biodistribution of silver NPs. 100 mg mL^−1^ of silver NPs was treated with cancer cells [70]. The presence of silver NPs in the mitochondria and nucleus was imaged by STEM; the direct effect of silver NPs in DNA damage and mitochondrial toxicity was confirmed. Peckys et al. applied the STEM method to provide information about the intracellular uptake of 30-nm AuNPs in live fibroblast cells (COS-7) in a microfluidic chamber [71]. Information about the distribution of AuNPs in pristine cells was obtained, since the cells are alive at the onset of STEM. That information includes the number of vesicles in the cluster, the vesicle diameter, and the number of detected AuNPs per 0.15 × 0.15 μm^2^ of the image.

#### 3.1.2. Atomic Force Microscopy

Atomic force microscopy (AFM) at a high resolution (below 10 nm) can provide information regarding topographical, mechanical, and chemical properties based on scanning a cantilever with a very sharp tip to measure interactions between a sample and a scanning probe tip [72]. AFM is appropriate for imaging soft/biological materials such as living cells and biomolecules in their natural environments, since they do not require a vacuum. Electron microscopes such as SEM and TEM are mostly suitable for imaging metallic NPs. For example, Chrzanowski and co-workers used infrared spectroscopy AFM (AFM-IR) to study the role of nanodiamonds (NDs) on fibronectin-FN as extracellular proteins and those proteins inside liver cells as intracellular proteins [73]. Conformational changes including protein aggregations and the formation of β-sheets, antiparallel β-turns, and intermolecular β-sheets can be observed for both intracellular and extracellular proteins at the ND-protein interface, implying an adverse effect of individual NPs on the protein structure. AFM also elucidates how curcumin-conjugated gold clusters (CUR-AuNCs) change the cell morphology of HeLa cancer cells [74]. The pore formation and surface roughness of cells are increased by CUR-AuNCs. Cancer cells treated with CUR-AuNCs indicate uneven, damaged surfaces at 24 h and breakage at 48 h (Figure 4c). This morphological damage in cancer cells confirms the antitumorigenic activity of the CUR-AuNCs, suggesting their potential for cancer therapy. Passeri et al., used contact resonance AFM (CR-AFM) as a nondestructive subsurface nanomechanical characterization tool for the detection of stiff (magnetic) NPs internalized in cells [75]. Magnetic NPs were internalized within microglial cells through a phagocytosis process. By increasing contact stiffness in the biological network, magnetic NPs agglomerate in the cell and show mechanical inhomogeneities that can be visualized via CR-AFM nanomechanical imaging. These results have been confirmed by other methods including light microscopy, HarmoniX^TM^, and magnetic force microscopy, revealing the profound implication of CR-AFM for the imaging of NPs into biological tissues.

#### 3.1.3. Confocal Laser Scanning Microscopy

Confocal laser scanning microscopy (CLSM) is an important class of optical imaging, enabling lateral (*xy*) and axial (*z*) images for the visualization of living specimens in a short time. CLSM can reject light from outside the focal plane (out-of-focus flare) through point scanning compared with other imaging procedures. This method has been applied exclusively in various types of cancer studies to clarify the internalization and distribution of NPs that affect cell morphology, resulting in toxicity. In a notable example, DNA-carrying NP-transported leukemia-targeting CAR genes into T-cell nuclei; they were rapidly internalized (120 min) within the cytoplasm, likely due to receptor-induced endocytosis [76]. Further research illustrates that two types of alginate NPs consisting of mannose-modified alginate (MAN-ALG) and an antigen model (ovalbumin (OVA) conjugate alginate (ALG=OVA)) were internalized into mouse bone marrow dendritic cells (BMDCs) (Figure 4d) [77]. The results indicate that MAN-ALG/ALG=OVA NPs enhance antigen uptake, delivery, and cytosolic release of the antigen in BMDCs. Xu et al. reported that the cellular uptake of coumarin-6 (C6) in 4T1 mammary tumor cells delivered via shrapnel NPs significantly increases with the use of matrix metalloproteinases (MMP-9) [78]. These results are promising for the inhibition of metastasis and the growth of breast cancer. CLSM is also able to provide nanoscale information on the transportation of NPs in intestinal epithelial cells and related mechanisms, enhancing the bioavailability of orally-administrated drugs. Both polymer and pure-drug NPs were employed to evaluate endocytosis and intracellular trafficking in Caco-2 epithelial cells [79,80]. CLSM colocalization and distribution methods illustrate the key role of actin in the internalization of polymer NPs. Endocytosis inhibitors lead to considerable suppression on the internalization of C6-polymer NPs. Intracellular trafficking increases when raising incubation time. CLSM further confirms that polymer NPs are transported to cells through both lysosome- and endoplasmic reticulum (ER)-trafficking pathways [79], whereas good colocalization with the ER has been reported for drug NPs [80]. These findings may shine new light on treatment of various types of cancers and provide important information for oncologists.

#### 3.1.4. Dark-Field Microscopy

Chan et al. utilized 3D dark-field microscopy based on NP scattering probes for three distinct biomedical applications: (i) the molecular imaging of whole tissues, (ii) the tracking of nanodrug carriers in mouse tumor models, and (iii) the characterization of lesions in a mouse model of multiple sclerosis [81]. 3D dark-field microscopy enabled the authors to observe the AuNP scattering signal in intestine and kidney tissues (Figure 4e). They also visualized both diffuse and localized signals from AuNPs in cleared tissues, regarding gold deposition. Dark-field imaging can also be used in combination with hyperspectral imaging that enables enhanced quantitative intensity for the detection, composition, and location of plasmonic NPs in biological samples. An example of hyperspectral dark-field microscopy (HSDFM) is the use of plasmonic nanoprobes to map and quantify multiple epigenetic marks at a single-modification resolution in single cells [82]. Plasmonic nanoprobes with strong LSP enable HSDFM to collect important information about cytosine modification in a single nucleus, including dynamic abundance, distribution, colocalization, and density. HSDFM can also shed light on the multiplex detection of proteins and oncogenic mRNAs for a single gene by spectrally differentiable AuNPs and AuNRs, [83]. The intracellular formation of gold particulates at a spatiotemporal resolution can be achieved with this single-cell optical clearing method.

#### 3.1.5. Light-Scattering Microscopy

Zhai et al. employed DLS to investigate the hydrodynamic diameter distributions and stability of non-crosslinked micelles (NCM) and core crosslinked micelles (CCM) that are remarkable in tumor combination chemotherapy [84]. Upon modification with fetal bovine serum, CCM shows a narrow size distribution and uniformity in the initial 30 min during the process, whereas the size of NCM increases to 176 nm, and then swells dramatically to over 600 nm with a broad distribution at 24 h, exhibiting the formation of large aggregates. Using DLS, the hydrodynamic radii (R_h_) of hexameric DNA and RNA nanocubes were 6.2 nm and 6.4 nm, respectively (Figure 4f) [85]. The DLS, cryogenic electron microscopy (cryo-EM), and native poly-acrylamide gel electrophoresis (PAGE) results confirm the formation of compact, closed molecules with different strands.

### 3.2. In Vitro Cell-Based Cytotoxicity Assay

In vitro biological assays aim to study the biological properties of NPs, identify their interferences and provide the first comprehensive insight into the potential sources of this interference. They also demonstrate the importance of the physicochemical characterization of NP formulations. There are various concerns regarding the safety of nanomaterials, particularly for biological applications. There has been a realization that further studies on nano-biocompatibility are needed. In vitro assays of material in any given matrix allow the examination of specific biological response(s) and/or mechanism of action under controlled conditions that are not easily studied in complex in vivo situations. In vitro assays are important, since they are cost-effective and provide quick results. There is no need to use animals in these assays, thus positively resolving ethical issues. Nevertheless, original physiological results (obtained through in vivo research) may differ from in vitro assays [49], since in vitro tests are not able to completely replicate the sophisticated interactions that take place in the body, mainly due to the use of one cell type or a mixture of several cells.

#### 3.2.1. Cell Viability Assays

Cell viability assays provide valuable information regarding the biocompatibility of materials by determining the number of survived or dead cells in a sample. It is thus vital in the assessment of nanomaterials for biomedical applications. A variety of cell types can be applied in cell viability assays, including but not limited to cancer cell lines, epithelial, fibroblast, hepatic, endothelial, neural, phagocytic, and red blood cells [86,87]. By using an MTT assay, Sokolova et al. showed that calcium phosphate NPs at low concentration (2.4 × 10^9^ mL^−1^) do not exhibit evidence of cytotoxicity on differentiated THP-1 cells after 1 h or 24 h of incubation [88]. The viability of THP-1 cells was about 50% when increasing the concentration of NP to 2.4 × 10^9^ mL^−1^. Such a cytotoxic effect for a higher concentration of NPs is possibly attributed to the increased concentration of PEI applied for the colloidal stabilization of the NPs. MTS is a negatively charged molecule converted into a soluble formazan product in the presence of phenazine methosulfate (PMS). Compared to MTT, this assay allows for the sequential analysis of treated cells with other assays. The content of tetrazolium salt can be measured photometrically at 490 nm. Similar to MTS, the WST-1 assay uses a negatively charged molecule that is a water-soluble tetrazolium salt. The WST-1 assay is more accurate than MTT for the evaluation of CNT toxicity, since insoluble MTT formazan crystals can bind to CNTs and produce inaccurate absorbance values [89]. WST-1 assays solve this problem by producing a water-soluble formazan product. ATP is linked to the metabolic activity of viable cells. It is thus employed as a marker to evaluate the viability and proliferation of cultured cells [90,91]. Its intracellular level is modified accurately in healthy cells due to the key role of ATP in cellular metabolism. This luminescence assay measures the metabolic activity of cells through the enzymatic transformation of intracellular ATP and D-luciferin into oxyluciferin and light.

#### 3.2.2. Cell Cytotoxicity Assays

Mean cytotoxicity/reactivity grades related to the concentration and toxicity of soluble components are diffused from the test sample. In vitro tests using primary cells can quickly screen for new chemical entities with serious toxicology ramifications. Nanomaterials have similar properties that are hazardous to other physical forms of the substance. Cell cytotoxicity assays are generally involved with cell death. (13 types of cell death mechanisms have been introduced) [92]. Cell death is a natural biological process that occurs as a result of an old cell dying or cell replacement, as well as localized injury and disease. There are three primary cell death mechanisms: necrosis, apoptosis, and autophagy. These are activated by nanomaterials [93]. Kostarelos and co-workers used a modified LDH assay to study the effect of different MWNTs on neuronal and glial cultures isolated from fetal rat frontal cortex (FCO) and striatum (ST) [94]. LDH results do not indicate any important reduction in either FCO and ST induced by MWNTs. Non-neuronal (mixed glial) cell cultures, however, were reduced in ST-derived mixed glial cultures due to the higher number of microglial cells in this brain region, as opposed to FCO-derived ones. These findings confirm that FCO-derived mixed glial cultures are less sensitive to MWNT exposure than ST-derived mixed glial cultures.

### 3.3. In Vivo Studies

In vivo experiments furnish information regarding biological interactions in living organisms through the use of animal models, including non-human primates. These experiments are the principal assays in assessing the toxicity of nanomaterials. They are able to determine the cytotoxicity and organotoxicity of NPs on a specific organism. Using this assay, the cellular uptake, distribution, metabolism, and clearance of nanomaterials can be obtained. They remain the most reliable methods to evaluate the toxicity of nanomaterials despite extensive time and cost. Typically, though, in vivo interaction of the biological system and NPs is sophisticated. In this section, we focus in detail on principal factors that affect the in vivo toxicity of nanomaterials.

#### 3.3.1. Blood Circulation

The initial meeting between the organisms and NPs occurs in the circulatory system; thus, the blood compatibility of NPs is fundamental for in vivo applications. The absence of blood compatibility may cause the adsorption of plasma proteins, platelet adhesion, and activation of complement cascades that lead to hemolysis and clot formation (thrombogenicity) [95]. Although blood compatibility is a prerequisite for NPs that are proposed for clinical applications, there few studies have evaluated the hematological and blood coagulation effects of NPs in vivo [96,97]. There are non-precise/standard procedures for the in vitro evaluation of these NPs. Hemolysis is the disintegration of red blood cells, given the changes in their cell membranes. During hemostasis, clot formation naturally occurs to stop bleeding from injured blood cells. Several negatively charged porous NPs (including MSNPs and zeolites) are able to accelerate the hemostasis of the blood by triggering a coagulation cascade. For example, Liu et al. developed ordered sphere-shaped MSNP doping with silver and calcium through one-step-based catalyzed self-assembly and subsequent ion-exchange for hemorrhage control [98]. These 3.2-nm NPs with a pore volume of 0.74 m^3^/g and a surface area of 919 m^2^/g showed suitable antibacterial and degradation abilities. The authors concluded that the optimal formulation of MSNPs is able to activate the intrinsic pathway of the coagulation cascade, cause platelet adherence, enhance blood clotting, and obtain hemorrhage control in rabbits using liver injury and femoral artery models. However, clot formation inside blood vessels through the injection of MSNPs may result in several unfavorable effects by blocking the blood vessels. This can cause strokes and death [99,100]. The use of generation-7 (G7) poly(amidoamine) dendrimers in zebrafish embryos, for example, adversely affects various blood components, generating rapid coagulation, which is notably different from normal coagulation mechanisms [101]. The interaction of positively charged dendrimers with negatively charged blood proteins (albumin and fibrinogen) causes coagulopathies (bleeding disorders). The clot formation of NPs should be examined prior to in vivo applications. Similarly, protein adsorption, usually regarded as the initial step in the interaction of NPs with blood, must be determined, since it leads to coagulation and other problems. Materials such as carbon and silver NPs trigger platelets and promote vascular thrombosis, as well as venous thrombus formation [102,103]. AgNPs (0.05–0.1 mg/kg i.v. or 5–10 mg/kg intratracheal administration) improve platelet aggregation and venous thrombus formation in rats. The results are consistent with in vitro analyses [103]. Carbon-based particles, including carbon black, CNT, and urban dusts, induce platelet aggregation and accelerate the rate of vascular thrombosis in rats [104]. Although all of the applied particles lead to the upregulation of GPIIb/IIIa in human platelets, dissimilar particles influence the release of platelet granules, as well as the activity of matrix metalloproteinase-, thromboxane- ADP, and protein kinase C-dependent pathways of aggregation.

Various analyses are used to evaluate plasma blood coagulation, such as thrombin time (TT), prothrombin time (PT), activated clotting time (ACT), and activated partial thromboplastin time (APTT), as well as the hematotoxicity properties of NPs, including complete blood count and possible genotoxic effects for clinical applications [105,106]. Ruiz et al. applied magnetite NPs with a narrow size distribution (mean size = 7 nm; SD < 0.15) functionalized with both PEG and dimercaptosuccinic acid (DMSA) to assess the hematotoxicity of these NPs using in vitro and in vivo tests [106].

#### 3.3.2. Pharmacokinetics

Pharmacokinetics (PK) demonstrate how organisms affect particular materials (e.g., nanostructures) after entering the body via several processes (e.g., absorption, distribution, metabolism, and elimination). Absorption occurs after administration via the interactions of nanomaterials with biological substances such as cells. They can then be distributed within numerous living organs, in which they can be altered or remain unchanged [107]. Nanomaterials may remain in an organ for an unspecified time before leaving to other cells or being excreted. A comprehensive analysis of PK is required to understand the toxicity of NPs and their potential effects in living organisms. Such biological information may influence the design/development of NPs with the highest efficiency in the field of biomedical areas. Table 1 represents a summary of PK studies of diverse NP types.

##### Absorption

NPs can be administrated into the body through three routes: the airways, skin, and through ingestion [123]. They can translocate to other organs, however only in small quantities [124]. Currently, in vivo results have been obtained mostly from intra-tracheal instillation and inhalation in rodents, including the bulk of toxicity information for diverse NPs [125]. Via the airways, NPs can access the body through the respiratory tract. The nasopharyngeal region absorbs particles of 10 to 20 nm [126]. The health effects of NPs are related to their residence time in the respiratory system and lung burden [127]. Reports show that smaller particles induce higher toxicity and lead to greater inflammatory reactions compared to larger particles of the same compounds [128]. Ultrafine TiO2 (<100 nm) and fine CNTs (<2.5 μm) at high concentrations cause tumors in the respiratory tract [129,130,131].

One of the major concerns about translocation is the ability of NPs to pass the air–blood barrier in the lungs, further allowing them to access other organs of the body [132]. Mucus and mucociliary escalators are defense mechanisms against particles in the body [133]. NPs successfully cross these barriers and translocate from the lung into the spleen, heart, liver, and perhaps other organs [134]. Gold particles introduced intra-tracheally into rats penetrated into the body by passing and translocating to other organs, including blood, urine, and different tissues, as well as being retained in the skeleton [135]. The translocation mechanisms of NPs via the air–blood barrier have not been clarified, although the importance of trans-cellular routes has been suggested [136]. The elimination of NPs from the lungs are maintained primarily through alveolar macrophages [137]. Inhaled NPs such as Ni, Mn, Co, and Cd NPs can also gain access to the nervous system via the olfactory nerves and/or the BBB [127]. Poorly soluble and insoluble NPs are able to enter the brain through the nasal route and to be uptaken via the BBB from systemic circulation [138].

NPs are capable of entering the human body via the skin through hair follicles or flexed or broken skin [125,139,140]. TiO2 NPs, for example, are often applied in commercially available sunscreen products. They may enter the body, but their translocation and dermal absorption are not completely understood and require further study [141]. Other NPs such as Qds [123] and fullerenes [142] enter the dermis [143,144], based on their size and surface coatings [145]. It is important to note that the skin penetration of NPs depends both on the properties of the NPs (e.g., size, dose, and surface charge) and skin conditions (e.g., skin barrier, age). In the case of Qds, the animal species plays a key role in the results. The skin of mice and porcine animals was penetrated by Qds, passing the dermal barrier; however, such penetration has not been reported for rat skin [143,146,147].

The gastrointestinal tract (GIT) is another route of administration of NPs into the body. Even though GIT exposure may lead to the translocation and uptake of NPs to various organs, there has been less research on this than on other routes. There are increasing concerns related to the possible entry of NPs into the blood stream via the GIT, mainly due to the increasing use of NPs in food packaging and processing or as food additives [148]. The kinetics of NPs in the GIT depend on the charge, in which negatively charged particles pass across the mucus layer, whereas positively charged latex particles stay in the negatively charged mucus [149]. Some diseases, for example, diabetes, may induce greater particle absorption in the GIT [149]. Although high concentrations of most NPs are quickly removed from the body through feces, small proportions are absorbed by the gut, showing size-dependent behavior [150,151].

##### Distribution

After the absorption of NPs into the body, they are distributed to diverse organs, cells, and tissues. Various factors influence the fate of nanomaterials in a human body, including their size, solubility, and surface charge. Optical imaging, post-mortem NP determination, and isotopic labeling with radioactive elements have been utilized to analyze the biodistribution of nanostructures [152,153,154]. Matsumoto et al. intravitally imaged the transport of fluorescent-labeled polymeric NPs (of 30 and 70 nm) at 10-min intervals to investigate their eruption and distribution in hypovascular human pancreatic BxPC3-GFP tumors implanted into BALB/c nu/nu mice by applying intravital CLSM [155]. They introduced dynamic vents and eruption as the time-limited formation of an opening in the vessel wall and vigorous extravasation through the vent. Eruptions occurred stochastically, although continuously, during the 10-h observation period. The occurrence of eruptions was observed for all modified groups in which 30-nm NPs indicated shorter-lived eruption plumes in contrast to 70-nm NPs. The size of the eruption plumes made by the 30-nm NPs was gradually smaller than those produced by 70-nm NPs. Vessel dynamics probably did not change according to NP size. The duration and eruption plume size were different due to the faster dispersion of smaller NPs. When NPs dispersed sufficiently, the fluorescence intensity decreased below the detection threshold. The use of 30-nm NPs rather than 70-nm NPs showed the quicker growth of background fluorescence, indicating that smaller NPs can be distributed more easily within a tumor. As the eruption begins during the early phase, the rate of plume growth in both sets reveals that smaller NPs are transferred more rapidly than larger ones. This is because, during the initiation phase, NP movement is essentially convection-driven, with bigger NPs experiencing slightly more movement resistance from the tissue structures. When complete, plume dispersal is controlled through diffusion. This is difficult for larger NPs. Therefore, vascular eruptions are responsible for the deep penetration of NPs, as well as restricted dispersion from the early plume. Campbell et al. provided mechanistic insights into NP–liver interactions in zebrafish at the molecular level, demonstrating an essential role of the stabilin-2 scavenger receptor (stab-2) in the uptake of anionic liposomes via sinusoidal (or scavenger) endothelial cells (SECs) [156]. They employed confocal microscopy to discern the real-time distribution of fluorescently labeled NPs in the bodies of zebrafish as a transparent model (Figure 5).

The authors used liposomes to evaluate the influence of surface charge on distribution. They intravenously injected three types of liposomes, including Myocet (neutral), EndoTAG-1 (positively charged), and Ambisome (negatively charged) into the duct of Cuvier of zebrafish embryos. Since each of these liposomes is formed from a combination of phospholipids and cholesterol, their cholesterol content and chemical structure (the level of saturation and the length of alkyl chains in fatty acids) are responsible for both physicochemical properties, thus explaining their in vivo fate. To decrease the possible variation of the chemical composition of liposome in relation to the distribution, the researchers also used 1,2-dioleoyl-*sn*-glycero-3-phosphocholine (DOPC), 1,2-distearoyl-*sn*-glycero-3-phosphocholine (DSPC), and 1,2-dioleoyl-*sn*-glycero-3-phospho-(1’-rac-glycerol) (DOPG). DOPG (a negatively charged, unsaturated lipid) and DSPC (a neutral saturated lipid) liposomes were strongly associated with venous ECs, whereas DOPC (a neutral, unsaturated lipid) circulated freely. NP-SEC interactions were inhibited by a competitive inhibitor of stab2, dextran sulfate, and other receptors. The authors finally applied these specific interactions for targeted intracellular drug delivery. This significant finding in zebrafish might well open a new avenue for researchers in biomedical engineering to solve the delivery issues of in vivo NPs.

##### Metabolism

The in vivo metabolism of NPs in tissues after systemic administration is a significant milestone for the rational design of nanostructure materials for clinical applications. This not only allows a better understanding of in vivo NP dynamics, but can also effectively minimize the toxicity risk of NPs. The degradation of NPs is essential for the successful delivery of therapeutics into clinical areas, since NP degradation is directly associated with the delivery and release of drugs at their desired location. Note that in vivo degradation of NPs is related to NP composition. For example, NPs made from Au, MSN, and QDs are generally not degraded in vivo, whereas other materials, such as Ag, CdSe, SPIONs, and ZnO, release metal ions [157]. Considering the case of SPIONs, several methods of analysis are frequently applied to quantify the degradation of NPs over time in the body, including inductively coupled plasma mass spectrometry (ICP-MS), temperature-dependent susceptibility measurements, electron paramagnetic resonance (EPR), MRI, and radioactive labeling [115,158,159]. Distinguishing between the NPs and their degradation products is fundamental. As an example, examining SPIONs in an ICP-MS test is not suitable due to the high amount of endogenous iron forms, whereas EPR provides information regarding the biotransformation of the superparamagnetic iron core in both the short and long term [160]. Gazeau et al. employed electron spin resonance (ESR) to quantify superparamagnetic iron oxide from the spleen and liver during a one-year period after intravenous injection in mice [161]. Gold/iron oxide nanoheterostructures were coated with PEG and an amphiphilic polymer, then they degraded in vivo. NPs coated with the amphiphilic polymer exhibited higher concentrations in the liver and spleen in contrast to PEG-coated NPs. Additionally, the initial coating showed a long-lasting influence on the degradation/clearance of NPs.

Apart from the quantification of NPs at the organ level, observing the NPs in the nanoscale in biological environments is indispensable for assessing the degradation of nanomaterials. Therefore, the use of high-resolution imaging techniques (probing the atomic structure of NP cores), as well as electron energy loss spectroscopy and energy dispersive X-ray (EDX) analysis, enables the local biodistribution of exogenous materials. These techniques have established that gold/iron oxide heterostructures and iron oxide nanocubes are degraded in the spleen and liver seven days after injection [161,162].

###### Clearance

From a toxicological perspective, nanomaterials should be cleared from the body after administration, because they may react with cells, organs, or tissues, which can cause severe health problems. The environment of the target tissues, as well as physicochemical properties including size, shape, coating, and surface charge, exhibit a significant impact on the clearance of nanomaterials from the body. There are three main pathways for the elimination of nanomaterials: hepatic, pulmonary, and renal clearance, which use the lungs, liver, and kidney organs, respectively, for removing nanomaterials.

Hepatic Clearance: The liver is the second-largest human organ after the skin and is a major organ for elimination of a wide range of drugs. Both the spleen and the liver are regarded as the main biological barriers for the translation of NPs, since they are responsible for sequestering most of the administrated NPs and prevent delivery to unhealthy organs. The liver consists of two lobes: parenchymal cells (i.e., hepatocytes) and nonparenchymal cells (i.e., sinusoidal endothelial cells, Kupffer cells, stellate, sinusoidal endothelial cells, and intrahepatic lymphocytes). These cells are concerned with the hepatic clearance of nanomaterials from the body [163]. The two lobes are separated by a middle hepatic vein. Blood is transferred to the liver through the portal vein and the hepatic artery. Hepatocytes and phagocytic Kupffer cells are important routes for the hepatic elimination of NPs. Several studies indicate that hepatic clearance is influenced by the physicochemical characteristics of nanomaterials. For example, larger NPs are cleared predominantly through the liver [164,165,166]. Typically, NPs bigger than ~200 nm are efficiently eliminated by Kupffer cells because of the slow blood flow in liver sinusoids. This provides enough time for the micropinocytosis and phagocytosis of NPs. However, smaller particles may pass through the endothelium into the Disse space or undergo transcytosis to be taken up by hepatocytes [167,168]. Hepatic clearance is also influenced by the surface charge of the NPs. For example, hepatocytes play a key role in the internalization of cationic NPs, whereas Kupffer cells are efficient for anionic NPs [168,169,170]. The in vivo clearance of nanomaterials is also associated with the interaction of NPs with biological components (i.e., enzymes, proteins, and biological fluids). A wide variety of phase I and phase II enzymes, including epoxide hydrolase, monooxygenase, esterases, and transferases, exists in the liver. This may affect the hepatic clearance of NPs [171].

Pulmonary clearance: Compared to the liver, the lungs are involved in the clearance of inhaled NPs that reach the body through the upper airways, since they are directly exposed to the environment. The fate of NPs is related to lung compartment deposition, which is size-dependent. For example, NPs smaller than 100 nm have a preference for deposition in the peripheral lung regions (i.e., alveoli), whereas larger particles with sizes of 1–10 μm are deposited in conducting airways with ciliated epithelial cells [172]. The alveolar macrophages (AMs) and airways are located at the front line of lung defense. The original mechanisms for the clearance of materials are mucociliary transport (as the most rapid elimination pathway for inhaled components) and phagocytic uptake (a slow process) [173]. Intranasally administrated NPs enter the areas of the alveoli in the lungs [174]. AMs phagocytize IONPs, digest them and their by-products, which are swallowed, or are released within pulmonary lymphatic vessels [175,176]. Research shows that the number of lung macrophages can grow in the presence of IONPs via the transportation of monocytes into the lung. This leads to the faster ingestion of NPs [177]. Other physicochemical properties of NPs, such as their coating and surface charges, can affect their clearance from the body. For example, Cho et al. utilized negatively charged cross-linked IONPs. The researchers showed the clearance of most of the NPs from the lung 3 h after penetration. This is because of increased macrophage uptake in the lung [178].

The clearance kinetics of nanosized drug carriers after pulmonary administration in the lung has not yet been fully elucidated. Kaminskas et al. compared lung clearance kinetics and the pulmonary pharmacokinetic pattern of two inhaled NPs (anionic liposomes and solid lipid NPs) with sizes of ~150 nm in rats. They used ^3^H-labeled structural lipids (phosphatidylcholine and tristearin) [179]. Although both liposomes and solid lipid NPs were deposited differently into the lung, they displayed similar clearance rates through the mucociliary escalator. Generally, however, prolonged lung exposure after an initial rapid clearance was seen for both NPs. Phospholipids from the liposomes were better absorbed from the lung, whereas the mucociliary clearance was more pronounced for solid lipid NPs. Lipid NPs were largely incorporated into the lung tissue and plasma proteins.

Renal clearance: The kidney plays a part in the effective elimination of foreign substances, such as nanomaterials, from the body. The main structural and functional parts of the kidney are nephrons, which are abundantly available. They assist the kidney to maintain the homeostasis of electrolytes and bodily fluids in extracellular, extravascular, and intracellular compartments. They are included in the specific filtering of proteins and saccharides from the blood, as well as NPs. NPs that are intravenously administrated can enter the blood vessels of the nephrons via the renal hilum and are then excreted in urine through the ureter, then by the urinary bladder [180]. Renal clearance is regarded as a suitable route for the clearance of NPs, because it requires minimal interaction and degradation within the body, causing fewer health hazards [181]. In this clearance route the NPs in the kidney first reach the glomerular capillary wall, which is the blood filtration site in the nephrons that is highly size-, shape-, and charge-dependent [182]. Other factors, for example, particle dose and molecular weight, strongly affect the renal clearance of NPs [183,184]. The glomerular capillary wall has three layers: the endothelium with fenestrae (70–90 nm), the glomerular basement membrane (2–8 nm pores), and the epithelium with filtration slits (4–11 nm) between the podocyte extensions [185]. Thus, only NPs with specific hydrodynamic sizes (below 6 nm) can be easily removed by the kidneys through the glomerular structure, whereas NPs with sizes above 6 nm are eliminated from the bloodstream via the RES (liver, spleen), requiring a much longer time [186,187,188]. Zheng et al. discovered that the glomerular wall in the sub-nanometer size range (~1 nm) acts as an atomically precise bandpass filter to notably slow the elimination of AuNCs of various sizes (Au_18_, Au_15_, and Au_10-11_) but with the same surface ligands (glutathione, GSH) (Figure 6) [189]. In Au_25_ (~1.0 nm), the renal elimination of AuNCs improves significantly through the additional number of atoms in a particle, since smaller AuNCs are more readily retained via the glomerular glycocalyx than larger particles. This interaction slows the extravasation of very small AuNCs from normal blood vessels and improves their accumulation in cancerous tissues through an enhanced permeability and retention (EPR) effect.

As opposed to small drug molecules, NPs are often passively accumulated with high concentrations for longer times due to EPR [190,191]. When larger NPs are compared with small, renal-clearable molecules [192,193], they accumulate in RES organs (the liver and spleen) [194], causing low targeting specificity and potential toxicity [195,196]. To resolve this problem for tumor targeting, NPs should be retained in the blood plasma at a relatively high concentration for >6 h; the sizes of NPs should be smaller than 6 nm, enabling them to successfully pass through kidney filtration [191,197]. For example, Zheng et al. compared the passive tumor targeting of two renal-clearable materials, organic fluorophores (IRDye 800CW) and ~2.5 nm glutathione-coated luminescent AuNPs (GS-AuNPs) [198]. In the initial stages of tumor targeting, AuNPs act like small molecules, with much longer tumor retention time than the dye, showing the EPR effect and effective renal clearance of NPs. The results revealed that GS-AuNPs are retained in blood plasma for a long clearance half-life of 8.5 h and are eliminated from normal tissues about trhee times faster than the dyes.

## 4. Physicochemical Properties of NPs Affecting Toxicity

The various physicochemical properties of NPs include their size, shape, surface functionality, surface charge, composition, hydrophobicity, aggregation, and solubility. They are essentially tied with their interactions with cells, thus influencing toxicity (Table 2). It is accepted that the control of these parameters should be optimum in biomedical/biological applications of NPs. In this section, we explain in detail the influence of these factors on the toxicity of nanomaterials.

### 4.1. Effect of Size

The size of NPs plays a critical role in their pharmacokinetics, therapeutic effect, in vivo biodistribution, and tumor accumulation [235,236,237]. NPs can passively accumulate in tumors based on their size due to the EPR effect and the different pore sizes of tumor vessels. Compared with healthy tissues, the EPR effect results in more than a 50-fold increase in the accumulation of NPs [238]. NPs are cleared via the mononuclear phagocytic cells in the RES of the spleen and liver, leading to inadequate accumulation at the target site. Another crucial barrier for drug delivery is the fast removal of NPs and drugs from the tumor tissue [239]. For example, the in vivo distributions of AgNPs with sizes of 20, 80, and 110 nm demonstrate that larger particles (80, 100 nm) are distributed mainly in the spleen, followed by the liver and the lung, whereas 20-nm NPs were primarily distributed in the liver, followed by the spleen and kidneys [240]. Several modifications have been used to enhance biodistribution and drug accumulation in tumors, such as the addition of a zwitterionic polymer membrane or folic acid- and dopamine-decorated hyaluronan to the surface of NPs [241,242,243]. Jiang et al. investigated the impact of size and surface chemistry on the accumulation of boronic acid-rich BSA NPs in tumors [244]. Particle residence time and drug accumulation were increased by adding the boronic acid group into the NPs. When doxorubicin accumulation at the tumor site reaches above 12% injected dose per gram of the tumor, a 16-fold drug is injected.

Concerted efforts have been made in cancer research to determine the optimum size of NPs, and to improve tumor treatment and diagnosis [245,246]. Tang et al. investigated the therapeutic effect and tumor accumulation of micelles with sizes between 30 and 200 nm. Smaller micelles indicate higher penetration and better therapeutic efficacy, whereas larger micelles show greater tumor accumulation [247]. The 12.1- and 27.3-nm PEG-coated AuNPs have stronger sensitization impacts than 4.8- and 46.6-nm particles via necrosis and apoptosis [248]. In vivo results show that all sizes of NPs reduce tumor weight and volume after 5 Gy radiation. Small NPs cause a considerable decrease in tumors compared to larger ones. Small NPs apparently accumulate with high concentrations at the tumor site. In vivo toxicity tests of blood biochemistry and immune responses confirm that PEG-coated NPs do not cause kidney and spleen damages, except for slight toxicity of the liver. Chen et al. tested [64Cu]-labeled perylene diimide (PDI) NPs with sizes of 30, 60, 100, and 200 nm as PTT agents, as well as photoacoustic (PA) and positron emission tomography (PET) imaging probes [249]. They found that 100 nm is an ideal NP size for differentiating popliteal and sciatic LNs. NPs migrate from popliteal LNs to sciatic LNs during an interval of about 60 min (Figure 7). PDI NPs of 60 nm are optimum for photothermal cancer therapy and tumor imaging because of their high tumor accumulation efficiency.

Small NPs can penetrate deeper into tumors; however, they are removed rapidly, with lower blood circulation time, whereas larger NPs cannot penetrate deeply into tumors and are retained in tumorous tissue for a longer time [250,251,252,253]. Size-reducible NPs based on temperature, pH, enzymes, and UV light as stimuli have been synthesized. They exhibited greater anti-tumor effects and better homogenous drug distribution in tumors [254,255]. There are, however, several factors that restrict the extravasation of large-to-small transformable NPs, comprising a dense tumor matrix, solid stress, and high interstitial fluid pressure (IFP). These factors weaken the effect of size-reducible NPs. Gao and co-workers designed a tumor microenvironment responsive and adjustable NP comprised of a laser-responsive nitric oxide (NO) donor, degradable HA, a small-sized dendrimeric prodrug (IDD) of DOX, and indocyanine green (ICG) as a photothermal agent [256]. HAase-induced size reduction occurred in a HAase-rich tumor microenvironment. This enabled the exposure of IDD with deep penetration. NPs improved the NO release to enhance the EPR effect in the presence of NIR laser irradiation and triggered potent hyperthermia for PTT. The Gao Group developed a size-reducible nanoplatform (AuNC@CBSA@HA) composed of degradable HA, cationic BSA (CBSA), and AuNCs for breast cancer and lung metastasis [257]. The HN shell protects the positive charge on AuNC@CBSA, thus reducing toxicity and increasing the circulating time of NPs. HA shields NPs with an active targeting ability that can be degraded in tumors to decrease the size of the NP, thus improving tumor penetration. Results show that 200 nm is optimal, with an ideal EPR effect when loaded with ICG and PTX to adjust the tumor microenvironment, drug delivery, and chemo-photothermal therapy. AuNC@CBSA-PTX-ICG@HA-NO_3_ with size-reducible properties displays a homogenous intra-tumor distribution. It shows high accumulation in breast cancer, impedes 88.4% of lung metastasis growth, and it prevents 95.3% of in situ tumor growth. The size of NPs can be regulated in the presence of different stimuli, changing from smaller to larger sizes in tumor microenvironments. Such an ability can combine the positive effects of both larger and smaller NPs, thus leading to optimal efficacy for tumor treatment.

### 4.2. Effect of Shape

The shape of NPs is also a critical factor that can significantly affect their role in biomedical applications. Although most laboratory-scale studies and clinical trials use spherical NPs, non-spherical particles have attracted attention in recent decades [258]. Such shapes as nanowires, nanorods, nanocubes, nanobelts, nanostars, nanothorns, etc., can be formed mainly via postsynthesis manipulation or using templates for polymeric NPs and wet chemical methods for metal NPs. In general, the shape of nanomaterials affects their biodistribution, biocompatibility, cellular uptake, and blood circulation [259]. Non-spherical NPs, such as discs and rods, have displayed higher tumor inhibition rates and cellular uptake efficacy. They can readily pass through the spleen [260]. Compared with other NP shapes, nanodiscs are internalized more effectively, and their mechanism of uptake efficiency, as well as their kinetics, display shape-dependent behavior [260]. There have been repeated efforts to study the bio-nano interactions of NPs. Elongated NPs possess better properties than spherical NPs [261]. The interaction, however, between the outcomes of suicide gene therapy and magnetic MSNPs (M-MSNPs) has not yet been clarified. Dong et al. addressed the efficiency of shaped-controlled M-MSNPs in magnetic hyperthermia therapy, MRI, and suicide gene therapy of hepatocellular carcinoma (HCC) [262]. Two different shapes of NPs, consisting of rod and sphere shapes, were fabricated by the Dong group. Rod NPs exhibited faster release behavior, greater loading capacity, better magnetically improved gene delivery, and higher magnetic hyperthermia properties. The magnetic ability of MSNPs has led to significantly efficient dual magnetically improved suicide gene therapy in vivo with the ability to observe therapeutic effects through MRI and with reduced toxicity. The author suggested the potential of both shaped M-MSNPs to be applied as safe and effective HCC theranostics. The Dong group yielded insight into the effect of the shape of M-MSNPs on the isolation and detection of circulating tumor cells (CTCs) [263]. They conjugated fluorescent M-MSNPs of rod and spherical shapes with the antibodies of EpCAM. They reportes good sensitivity of both NPs on CTCs. Rod M-MSNPs offer higher effective detection and faster enrichment of CTCs in real clinical blood samples and spiked cells, as opposed to their sphere-shaped counterparts. Recently, molecular bottlebrushes as unimolecular micelles in the shape of spheres, rods, and worms were synthesized by means of click chemistry and controlled/living polymerization for cancer therapy [264]. IR780 as a photothermal agent was loaded into these core-shell NPs. The maximum loading content of ca. 25% did not result in aggregation or morphological changes of these molecular NPs in cell culture media. Unimolecular rod NPs exhibited the best penetration in vitro and in vivo, as well as cellular uptake that efficiently inhibited tumors. Although shape notably affected cellular uptake, circulation time, tumor accumulation, etc., it is not clear how NP shape influences loading capacity and drug release. Rampersaud et al. compared the drug release and efficiency of solid spherical and cage shapes of IONP-loaded riluzoles as anticancer drugs [265]. The cytotoxic effect of nanocages was two times greater, and the drug delivery of spherical NPs was three times less effective than their counterparts. This difference in drug delivery can be attributed to the impact of charge screening on the NP surface. When loaded with NPs, the charge of drugs can affect the overall charges of nanostructures (NP-loaded drugs). Thus, the surface area of NPs plays an important role in the way NP shape influences drug release and efficiency.

### 4.3. Effect of Surface Charge

Surface charge can substantially influence the efficiency and the in vivo fate of NPs [266,267]. The surface densities and charges (negative, neutral, or positive) are different for each NP. Typically, neutral and negatively charged NPs exhibit higher biocompatibility and longer circulation half-lives, as well as enhanced stealth-like effects, and reduced undesirable clearance [268,269]. Due to the negative charge of cell membranes, both neutral and negatively charged NPs display lower degrees of internalization compared to their positively charged (cationic) counterparts [241]. Cationic NPs interact with negatively charged groups on the cell surface, and they are transferred through the cell membrane. The degree of positive charge is related to cellular internalization [270]. Cationic polymers, including PEI, DEAE-dex, poly(L-lysine) (PLL), and chitosan, have been applied in cellular uptake. The highest density of charged groups has been reported for PEI, which significantly enhances membrane permeability [271,272]. Several cationic materials without fusogenic activity can disrupt endosomal membranes via their buffering capacity [273]. Plasma proteins can interact with the positive surfaces of NPs and improve cellular uptake [274]. Cationic NPs are apparently important in mucosal drug delivery, improving in vivo association to the cornea, nasal mucosae, and GIT mucosae [273]. They apparently pass the BBB to a higher extent, thus improving brain penetration [275]. Cationic NPs, however, are not essential for effective endocytosis. Neutral or negatively charged NPs indicate effective cellular uptake when conjugated with targeting ligands. Dante et al. reported the quick localization of negatively charged magnetic and cadmium-based semiconductor NPs (administered at low concentrations of 10 nM) on neuronal membranes. This causes electrophysiological manipulations (Figure 8) [276].

The same neutral or positively charged NPs display low or no nonspecific interaction with the cellular membrane. The interaction of negative NPs is apparently selective for excitable neuronal cells and is dependent upon neuronal activity. The results of the pharmacological alternation of electrical neuronal activity suggest the impact of neuronal activity on the neuron–NP interaction. The role of different charged NPs on in vitro and in vivo toxicity may be different. For example, negatively charged Qds have been reported to be considerably less cytotoxic than cationic Qds [277]. Although highly toxic cationic Qds cause alternations in genes related to mitochondrial functions, negatively charged Qds improve the gene expression of those related to DNA damage and proinflammatory cytokines.

### 4.4. Effect of Surface Functionality

The presence of functional groups on the surface of NPs can surprisingly improve their functionality and biomedical applications. Compared to pristine nanomaterials, functionalized NPs can achieve better biocompatibility, as well as more effective targeting and delivery by adjusting their functional groups. Such properties can be obtained via the utilization of various coating approaches, such as PEGylation [278], ligand exchange [279], and targeting-moiety conjugation [280]. These approaches can significantly enhance the properties of NPs and result in higher bioderived recognition, circulation time, and weakened nonspecific interactions. PEGylation is the attachment of PEG to the surface of NPs, where ethylene glycol units form potent connections with water molecules, leading to the formation of a hydrating layer [281]. This hydrating layer prevents protein adsorption following elimination through the mononuclear phagocyte system (MPS), which can enhance the circulation time of NPs. One example is increasing the circulation time of the drug from minutes to hours via the PEGylation of liposomal doxorubicin [282]. PEG is regarded as the most desirable polymer for the functionalization of NPs, as opposed to polysaccharides, poly(amino acid)s, poloxamers, and polyvinyl alcohols with the same coating properties [283].

The strategy of functionalizing NPs with biological molecules can induce a more therapeutic impact with the delivered drugs after intracellular uptake. The best example of using NPs functionalized with biological molecules is the use of D-α-tocopheryl derivatives in cancer therapy [284,285]. D-α-tocopheryl polyethylene glycol-succinate (TPGS) and D-α-tocopheryl succinate (TOS) are of particular interest, since they are not toxic against normal cells and present a highly selective antitumor effect by destabilizing cancer cell mitochondria [286,287]. TOS-functionalized materials display great in vivo circulation properties and high drug loading capacities [288,289]. Qi et al. used penetrating peptide tLyP-1-functionalized NPs (tLPTS/HATS NPs) constructed from two modularized amphiphilic conjugates of TOS-grafted hyaluronic acid (HATS) and tLyP-1-PEG-TOS (tLPTS) for targeted tumor therapy [290]. The resultant NPs, with a mean diameter of 110 nm, showed excellent encapsulation efficiency (93%) and sustained drug release. In in vitro experiments, the tLPTS/HATS NPs displayed much higher cytotoxicity, anti-invasion ability, and apoptosis toward both MDA-MB-231 and PC-3 cells, and they exhibited better intracellular delivery than HATS NPs. tLPTS/HATS NPs exhibit excellent tumor penetrating and inhibitory performance on both PC-3 a and MDA-MB-231 tumor spheroids. Better distribution and lasting accumulation throughout tumor regions, along with lower systemic in vivo toxicity and more therapeutic efficacy, in mouse models, have also been observed for functionalized NPs.

The functionalization of NPs with peptides can decidedly enhance their properties, especially for tumor therapy [291]. For example, Kinnari et al. reported the fabrication of thermally hydrocarbonized porous silicon NPs (Si NPs) functionalized with a tumor-homing peptide, the subcutaneous mammary-derived growth inhibitor (MDGI) [292]. In contrast to pristine NPs, functionalized NPs display efficient targeting and approximately 9-fold greater accumulation in the tumor site, whereas no important accumulation affecting other vital organs has been reported for either NP. NPs can also show active targeting abilities based on receptor-mediated endocytosis when functionalized with cell-recognizable targeting ligands such as low-molecular-weight materials (e.g., folate) [293], monoclonal antibodies [294], and endogenous targeting peptides [295]. Pep-1, as a specific ligand of interleukin 13 receptor α2 (IL-13Rα2) with high specificity and affinity, can interact with IL-13Rα2, thereby passing the blood–tumor barrier (BTB) and homing in on the glioma [296]. Based on IL-13Rα2-mediated endocytosis, Xin et al. functionalized the surface of PEG-PLGA NPs with Pep-1 using a maleimide-thiol coupling reaction for glioma-targeting delivery [297]. Enhanced penetration in 3D avascular coumarin-6 (C6) glioma spheroids, as well as a highly improved cellular association in rat C6 glioma cells, were observed for the functionalized NPs. Deeper tumor penetration, better extensive fluorescence intensity, and tissue biodistribution were monitored for the C6 tumor spheroids treated with Pep-NP (Figure 9).

### 4.5. Effect of Hydrophobicity

The hydrophobicity of NPs influences their nanobiological interactions, membrane damage, and toxicity [230,298]. The behavior of hydrophobic NPs is attributed to their excellent ability to react with cell membranes, specifically to their affinity to the hydrophobic cores of the bilayer and membranes. They can also cause an immune response through the activation of immune cells [299]. Hydrophobic NPs are available in smoke, soot, dust, and volcanic ash and are mobile in aqueous solutions due to their small size. The mobility and reactivity of hydrophobic NPs raise serious concerns regarding their biomedical applications. Small hydrophobic NPs of <10 nm in diameter can penetrate and accumulate in bilayer cores [300,301]. Generally it is thought that these small NPs cannot escape from bilayer cores after they are trapped in them, whereas larger NPs can only penetrate cells via slow energy-dependent processes, namely, endocytosis, lasting from seconds to minutes [302]. Small ions, molecules [303,304], hormones, and fullerenes [305,306] are able to translocate directly through the lipid bilayer via passive diffusion. This process takes only hundreds of nanoseconds and is energy-independent [303]. As a consequence, this considerable difference in the internalization time of particles can significantly affect cell responses to NPs. Different approaches, often based on numerical simulations, have been suggested to translocate small NPs, i.e., altering NPs’ coatings through disposable ligands [307], peptides [308], stripped nanopatterns with a controllable symmetry [270,309], or changing the NPs’ shapes [310]. None of these NPs have been able to translocate experimentally through a phospholipid bilayer. Some can internalize within cell membranes. Baulin et al. reported the direct translocation of hydrophobic NPs through lipid bilayers, both theoretically and experimentally, within milliseconds [311]. AuNPs with larger diameters of 5 nm were internalized and formed pores in the bilayer, as opposed to those with diameters smaller than 5 nm, which were trapped in the bilayer. NPs translocate only once because of the exchange of the coating lipids between the lipid bilayer and the NPs. Thus, the induction of lipid membrane permeability can be markedly affected by NP surface hydrophobicity. Rotello et al. employed AuNPs with different surface hydrophobicities but similar surface charges and sizes by human serum concentration (10% and 50%) and terminal functional groups [312]. In contrast to 50% serum, increasing the NP surface hydrophobicity in the presence of 10% serum was found to lower the abundance of apolipoproteins. Increasing the NP hydrophobicity with 10% serum led to a quantitative reduction in coronal immunoglobulins, although the opposite behavior was monitored in 50% serum. This study shows the importance of bifunctional linkers in adjusting the hydrophobicity and charge of AuNPs, which can decrease the protein corona and improve cellular uptake [313].

The hydrophobicity of NPs plays a key role in their interactions with biological barriers. For example, Porret et al. evaluated the interaction of a series of AuNCs (metal core ~1–1.5 nm) with (i) serum in solution, (ii) a lipid bilayer model system integrated into a microfluidic device, and (iii) different cell types (A375 melanoma and U87MG human primary glioblastoma cells) [314]. Controlling the hydrophilicity/hydrophobicity balance on the surfaces of NPs can inhibit the production of a corona and can enhance colloidal stability in a serum-containing medium. Higher lipid bilayer membrane insertion and more rapid cellular uptake were obtained by increasing the surface hydrophobicity of the AuNCs. The presence of a hydrophobicity threshold leads to colloidal instability, lipid bilayer damage, and cytotoxicity. The metal-ligand shell hydrophobicity reportedly has a crucial impact on the fluorescence signal of the AuNCs. The hydrophilicity of NPs increases blood circulation; however, it prevents cellular uptake and inhibits NP interactions with tumor cells [315,316]. Yang et al. designed poly(N-isopropylacrylamide) (PNIPAM)-based nanogels with rapid adaptive hydrophobicity for cancer therapy to address these issues, along with achieving higher tumor penetration (Figure 10) [317]. The nanogels were synthesized via NIPAM, with sulfobetaine methacrylate (SBMA) as comonomers with pH-responsive N-methylallylamine (MAA) in the presence of the disulfide bond-containing N, N’-bis(acryloyl) cystamine (BAC) as a crosslinker. The nanogels became hydrophilic in the blood and quickly switched to hydrophobic at the tumor site due to the acidic environment of the tumor, leading to higher blood circulation, deeper tumor penetration, and effective uptake by bulk tumor and cancer stem cells. Combined with a redox-responsive and lysosomal pH-regulated charge reversal drug release, the nanogels escaped from lysosomes and released DOX. As a result, the nanogels not only enhanced in vivo anti-cancer efficacy but also reduced the side effects of DOX. The ratio of cancer stem cells decreased after the nanogel-DOX treatment.

### 4.6. Effect of Aggregation

Compared with parameters such as shape, size, and surface chemistry, the aggregation effects of NPs have been studied much less. Due to exposure to biomolecules such as proteins and ions, NPs can aggregate in cell culture environments that can markedly increase the risk of toxicological side effects in biomedical applications [318]. Aggregation happens when van der Waals attractive forces between particles are higher than the electrostatic repulsive forces generated by the NP surface [319,320]. The high content of ions in biological media can reduce the screening length of charged groups on the NP surface. The high concentration of proteins finally leads to a thermodynamically preferred replacement of surface-associated molecules with serum proteins [321]. This behavior results in destabilization of the NP surface and induces a population of well-characterized NPs dispersed in a buffered solution to aggregate in a biological environment, such as cell culture, blood, lung surfactant, or saliva. One apparent solution is to coat NPs with a stabilizing shell that does not aggregate in a short period, although NPs are eventually aggregated in both in vitro and in vivo studies [322,323]. Albanesen and Chan show the role of transferrin-coated AuNP aggregates with different sizes on cell uptake and toxicity [324]. The aggregation of NPs in HeLa and A549 cells caused a 25% decrease in uptake compared to single and monodisperse NPs. The largest synthesized aggregates showed a 2-fold increase in MDA-MB 435 cell uptake. These findings indicate an important influence of interactions and cell type on the cellular uptake of NPs.

The aggregation of NPs is a thermodynamics-driven phenomenon, which has been extensively investigated as a function of pH, time, humic acid adsorption, and ionic strength [325,326]. The aggregation of NPs in the presence of simple electrolytes has been satisfactorily described by the Derjaguin–Landau–Verwey–Overbeek (DLVO) theory of colloid stability, according to which aggregation via screening causes surface charge and depends on repulsive electrostatic and attractive van der Waals forces [327]. Among the physicochemical parameters of NPs, the aggregation tendency is critical for in vitro and in vivo cellular responses [8]. Hahn et al. demonstrated the dissolution behavior and aggregation tendency of nanostructured ZnO in RAW 264.7 murine macrophages as a function of concentration in phosphate-buffered saline (PBS, pH 7.4) [328]. Smaller aggregates of NPs are more effective than larger aggregates in inducing mitochondrial dysfunction, producing elevated intracellular ROS, and causing apoptosis in RAW 264.7 cells. Hence, the concentration of NPs considerably determines the aggregation and dissolution tendency, which further influences their biomedical applications. AuNPs generally aggregate upon introduction into high-salt or high-protein media [329]. Such aggregation is formed due to the presence of crosslinking species (e.g., polyelectrolytes or divalent cations) as a function of pH [330]. Murphy et al. describe the formation of protein corona, which can serve as an efficient approach to avoid AuNP aggregation in cell media. They investigated the effect of NP aggregation on human dermal fibroblast (HDF) cells [331]. Aggregation can be avoided through the addition of AuNPs to fetal bovine serum (FBS) and then to a buffer. Coating AuNPs with lipids avoids the aggregation of NPs. This aggregation affects the cellular uptake and toxicity of particles, in which the uptake of aggregated citrate AuNPs can be three times greater than non-aggregated anionic citrate AuNPs. The toxicity of non-aggregated cationic AuNPs to HDF cells is four-fold lower than that of aggregated cationic AuNPs. Actin fiber disruption is influenced more than aggregated AuNPs.

The efficient accumulation of NPs in tumors is of great importance for cancer diagnosis and treatment. It has been hypothesized that the retention and cellular uptake of small NPs in tumors can be improved through the stimulated aggregation of NPs in tumor sites through the tumor microenvironment [332]. Based on this hypothesis, Ji et al. fabricated smart AuNPs of 16 nm via surface modification with mixed-charge zwitterionic self-assembled monolayers to assess the role of aggregation on the accumulation of NPs in tumors (Figure 11). The prepared NPs were stable at the pH of normal tissues and blood, whereas they aggregated quickly in response to the acidic extracellular pH of solid tumors. Zwitterionic AuNPs show ultrasensitive, quick, and reversible responses to the pH change between pH 7.4 and pH 6.5. This feature allows the AuNPs to be properly dispersed at pH 7.4 with great stealth ability but instantly aggregating at pH 6.5, resulting in highly improved uptake by cancer cells. Zwitterionic AuNPs exhibit a blood half-life with greater tumor accumulation, retention, and cellular internalization compared to nonsensitive PEGylated NPs. Given the results of photothermal ablation, the aggregation of AuNPs can be used for cancer NIR photothermal therapy. Such findings suggest that controlling the aggregation of NPs responsive to aspects of the tumor microenvironment, such as matrix metalloproteinase enzymatic activity, low pH, or low O_2_, can improve the cell uptake and retention of inorganic NPs in tumors.

### 4.7. Effect of Solubility

Nanomaterials can be classified as soluble or insoluble NPs in various biological media. This solubility feature may increase the accumulation of NPs in tissues or other biological environments and lead to toxicity [278]. After NPs are partially or completely dissolved, their structures break down into smaller parts. A high or even a low degree of solubility of NPs may engender a wide range of problems, thus limiting or precluding their biomedical application. Compared with soluble NPs, insoluble particles are able to stay in a person’s lungs for an extended period of time, thus causing severe adverse health effects [333]. Therefore, understanding the solubility of NPs will help to facilitate the development of safer materials in nanotechnology-based biomedical applications.

## 5. Future Prospects

Nanostructured systems can be used in the fabrication of novel bio-products for use in a variety of applications. However, the deployment of such materials can pose health hazards. Therefore, there is a need to acquire knowledge about their potential risks when they are used in nanotechnology, biotechnology, electronics, and medicine. Future developments will very much depend on a better understanding of the health and safety risks associated with the use of these materials.

Marketing nanomaterials has developed rapidly. To keep up with this demand, it is therefore necessary to design appropriate educational courses and provide consultancy services on a variety of aspects, including strategic decision making, commercialization lifecycles, and risk management, to support decision making. Government, manufacturers, nano-research laboratories, investors, lawyers, and professional media should make use of these opportunities in order to upskill their workforce. Private nano-companies must provide education, courses, and consulting services for nano-research laboratories aimed especially at nano-toxicological issues. Using standard services, one can accomplish enormous cost savings and reduce risk by proactively addressing possible safety concerns.

## Figures and Tables

**Figure 1 pharmaceutics-13-01615-f001:**
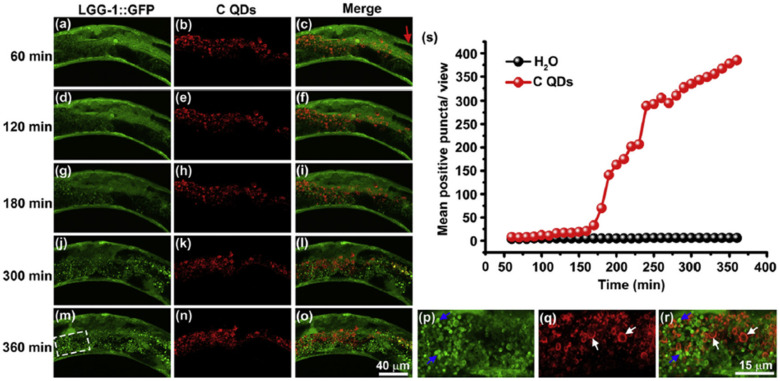
Real-time tracking of autophagosome formation in intestinal cells of C-QD-injected worms. (**a**–**c**), (**d**–**f**), (**g**–**i**), (**j**–**l**), and (**m**–**o**) are the fluorescent microscopy images of LGG-1::GFP and C QDs in intestinal cells of a C-QD-injected worm at 60 min, 120 min, 180 min, 300 min, and 360 min post-microinjection, respectively; (**p**–**r**) present enlarged views of the boxed areas in (**m**–**o**); blue arrows indicate the aggregated LGG-1::GFP; white arrows point to the internalized QDs; the red arrow in (**c**) indicates the injection site in the intestinal cells; (**s**) quantitative analysis of the aggregated LGG-1::GFP puncta subjected to C QDs or H_2_O treatment over 6 h. LGG-1 is the worm ortholog of the vacuolar protein Atg8/MAP-LC3). Reproduced with permission from [31], Elsevier, 2015.

**Figure 2 pharmaceutics-13-01615-f002:**
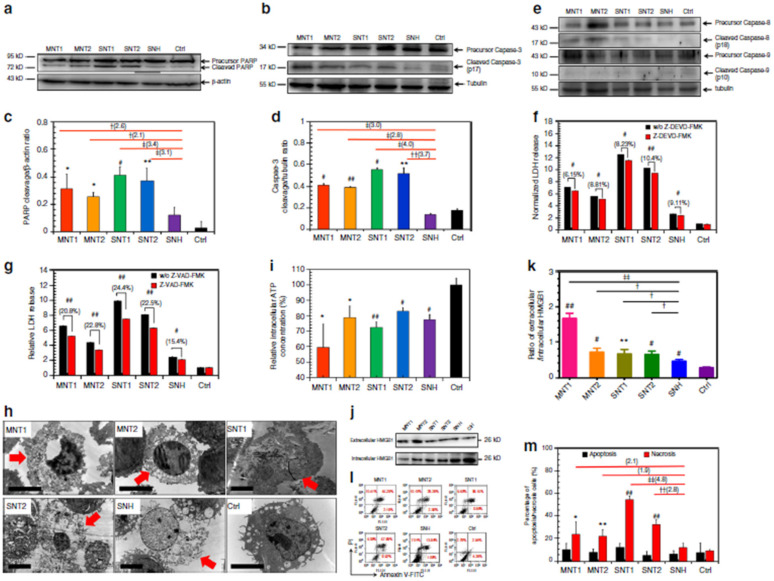
SNH induced less apoptosis and necrosis than CNT. (**a**,**b**) Western blot analyses of (**a**) PARP and (**b**) caspase-3 cleavages after different nanocarbon incubations. (**c**,**d**) Quantitative cleavage ratio measurements of (**c**) PARP and (**d**) caspase-3 in nanocarbon-incubated cells according to the integrated optic density (IOD) value, detected based on Western blot imaging (*n* = 3). (**e**) Western blot analyses of caspase-8 and caspase-9 cleavages after nanocarbon incubations. (**f**,**g**) Cytotoxicity detection of different nanocarbons with and without two caspase inhibitors, (**f**) Z-DEVD-FMK and (**g**) Z-VAD-FMK) (*n* = 4). (**h**) Transmission electron microscopy images of dead cells caused by different nanocarbons. Red arrows show the typical necrosis characteristics of cells. Scale bar: 5 μm. (**i**) Intracellular ATP detection after nanocarbon incubations (*n* = 4). (**j**) Immunoblot analysis of extracellular and intracellular HMGB1 after cellular incubations with different nanocarbons. (**k**) Quantitative ratio of extracellular HMGB1 to intracellular HMGB1 according to the IOD detection based on WB imaging (*n* = 3). (**l**) Flow cytometry analysis of cells based on the Annexin V/PI assay after nanocarbon incubations. (**m**) Quantitative comparison of apoptosis and necrosis caused by different nanocarbons detected using an apoptosis/necrosis assay kit (*n* = 4). In (**c**,**d**,**f**,**g**,**i**,**k**,**m**), data are presented as means ± s.d. Statistical significances were calculated by Student’s *t*-test. In (**c**,**d**,**k**), and (**m**), data were compared with control (Ctrl) and SNH groups separately. Versus Ctrl: * *p* < 0.05, ** *p* < 0.01, # *p* < 0.005, ## *p* < 0.001. Versus SNH: † *p* < 0.05, †† *p* < 0.01, ‡ *p* < 0.005, ‡‡ *p* < 0.001. The values in brackets denote the data ratios compared to SNH group. In f, g, data were compared with no-inhibitor-added groups for each type of nanocarbons: # *p* < 0.005; ## *p* < 0.001. Reproduced with permission from [34], Nature, 2018. Abbreviations: PARP, poly-ADP-ribose polymerase; Z-DEVD-FMK, specific caspase-3 inhibitor; Z-VAD-FMK, pan-caspase inhibitor; MNT, multi-walled carbon nanotubes, SNT, single-walled carbon nanotubes.

**Figure 3 pharmaceutics-13-01615-f003:**
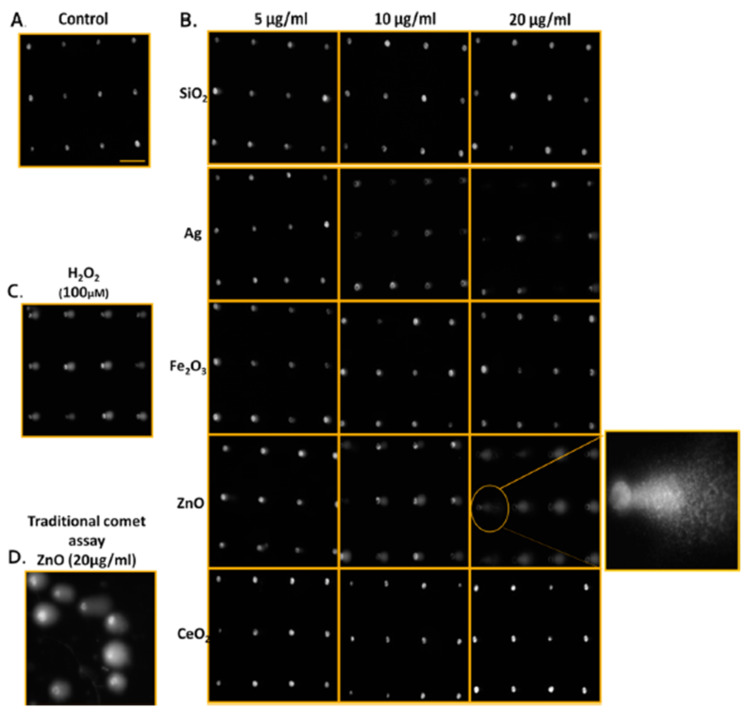
Qualitative images of nanoparticle-mediated ENP DNA damage in TK6 cells using both CometChip and standard comet assays. (**A**) Media-treated control cells. (**B**) TK6 cells were exposed to industrially relevant ENPs at concentrations of 5, 10, and 20 μg/ml for 4 h and evaluated using CometChip technology. The expanded view illustrates the morphology of the comet structure induced from 4 h exposure of zinc oxide ENP in TK6, revealing significant DNA damage. (**C**) Positive control cells treated with H2O2 (100 μM) for 20 min. (**D**) Traditional comet assay of TK6 cells treated with ZnO (20 μg/ml) for 4 h for comparison to CometChip qualitative assessments. Horizontal scale bar represents 100 μm. Reproduced with permission from [52], American Chemical Society, 2014.

**Figure 4 pharmaceutics-13-01615-f004:**
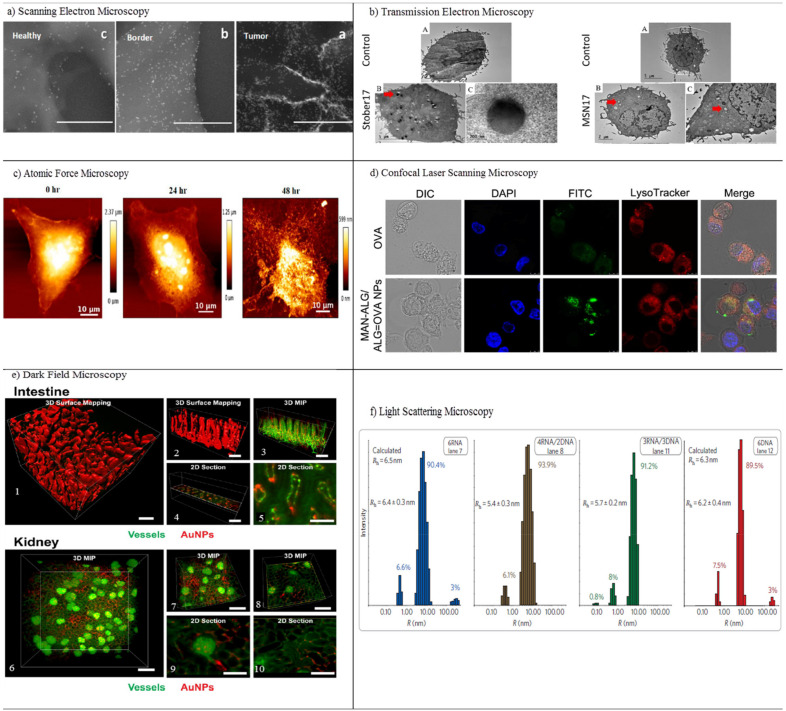
In vitro characterization methods. (**a**) Scanning electron microscopy (SEM) images of the different sites (a–c) in the oral SCC slide, in a nanometric resolution. The GNRs appear as bright rods. The nanoparticles’ concentration gradually decreases from the tumor to the healthy sites. Scale bar is 1 μm. Reproduced with permission from [66], American Chemical Society, 2016. (**b**) Transmission electron microscopy (TEM) of RAW264.7 cells treated with media (control) or treated with different concentrations of porous and nonporous SNPs for 4 h. The SNPs were taken up by cells and localized inside vesicles. The dose-dependent increase of the cellular association of SNPs in both types of nanoparticles is visualized. Red arrows indicate particles inside cells. Particles were not observed inside the nucleus. Reproduced with permission from [69], Elsevier, 2017. (**c**) Atomic force microscopy (AFM) imaging of CUR-AuNCs treated with HeLa cells in different intervals of time, 0, 24, and 48 h. Reproduced with permission from [74], American Chemical Society, 2018. (**d**) Confocal laser scanning microscopy (CLSM) imaging of DCs after incubation with free OVA and MAN-ALG/ALG=OVA NPs. Reproduced with permission from [77], Elsevier, 2017. (**e**) 3D dark field microscopy of AuNPs in mouse intestine and kidney tissues, with (1) surface mapping of the 3D image of intestinal tissue containing 50-nm AuNPs, showing the morphology of villi. (2) Smaller segment of the image from (1). (3) 3D maximum intensity projection (MIP) of the same region of intestinal tissue showing the arrangement of blood vessels and the distribution of AuNPs. (4) Position of the 2D section in (5) showing the distribution of AuNPs within a single villus. (6, 7) 3D maximum intensity projection of blood vessels and AuNPs within kidney tissue with brightly stained glomeruli visible. (8) Location of 2D sections of (9) and (10) showing the local distribution of AuNPs within and around a glomerulus. Scale bars indicate 200 µm for (1), (2), (3), (4), (6), (7), (8) and 100 μm for (5), (9), (10). Reproduced with permission from [81], Royal Society of Chemistry, 2017. (**f**) Light scattering microscopy of size histograms for 6-stranded cubes. Reproduced with permission from [85], Nature, 2010.

**Figure 5 pharmaceutics-13-01615-f005:**
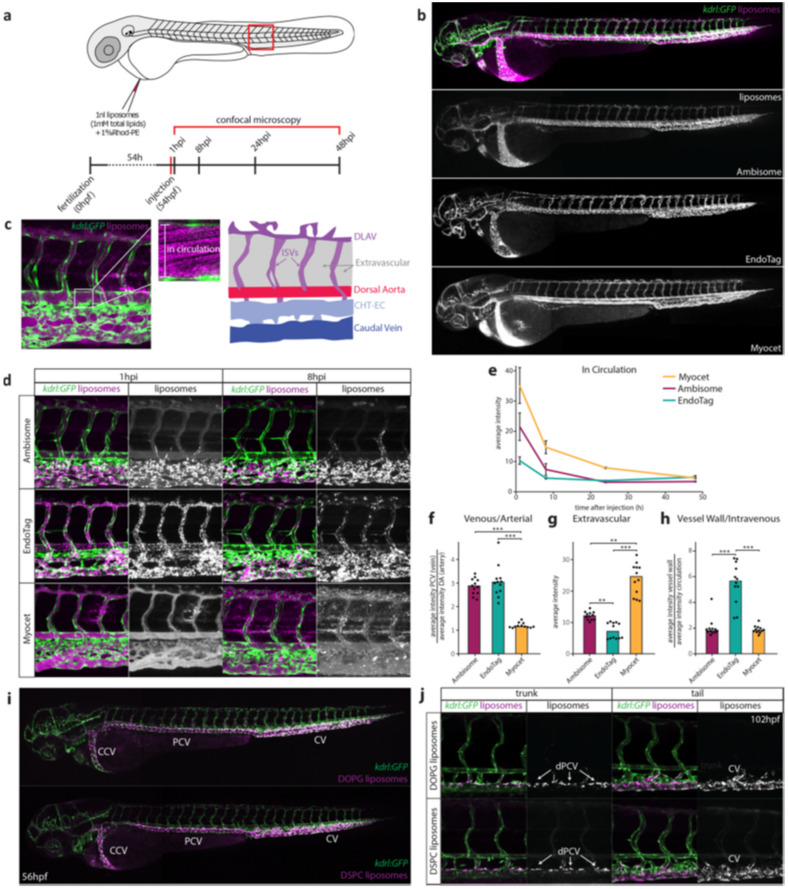
A zebrafish model for liposome biodistribution. (**a**) Schematic of liposome injection and quantification in zebrafish. Fluorescently labeled liposomes (1 mM total lipids containing 1 mol% Rhod-PE) were injected into the duct of Cuvier at 54 hpf. Confocal microscopy was performed in a defined region (boxed) caudal to the yolk extension at 1, 8, 24, and 48 h after injection. (**b**) Whole-embryo view of liposome distribution in kdrl:GFP transgenic embryos, 1 hpi with three different liposome formulations (AmBisome, EndoTAG-1, and Myocet). (**c**) High-resolution imaging allows the quantification of liposomes in the circulation (measured in the lumen of the dorsal aorta (white box)) and liposome association with different blood vessel types. CHT-EC: caudal hematopoietic tissue endothelial cells, DLAV: dorsal longitudinal anastomotic vessel. ISV: intersegmental vessel. (**d**) Tissue=level view of liposome distribution in kdrl:gfp transgenic embryos, 1 h and 8 h after injection, with three different liposome formulations and a single confocal section through the dorsal aorta (DA) at 1 h after injection. (**e**) Quantification of liposome levels in circulation based on mean rhodamine fluorescence intensity in the lumen of the dorsal aorta at 1, 8, 24, and 48 h after injection (error bars: standard deviation.) *n* = 6 individually injected embryos per formulation per time point (in two experiments). (**f**) Quantification of liposome levels associated with venous vs. arterial endothelial cells based on rhodamine fluorescence intensity, associated with caudal vein (CV) vs. DA at 8 h after injection. (**g**) Quantification of extravascular liposome levels based on rhodamine fluorescence intensity outside of the vasculature between the DLAV and DA at 8 h after injection. (**h**) Quantification of liposome levels associated with the vessel wall based on rhodamine fluorescence intensity, associated with all endothelial cells relative to rhodamine fluorescence intensity in circulation at 1 h after injection. (**f**−**h**) Bar height represents median values, dots represent individual data points, and brackets indicate significantly different values (*: *p* < 0.05, **: *p* < 0.01, ***: *p* < 0.001), based on Kruskal−Wallis and Dunn’s tests with the Bonferroni correction for multiple testing. n = 12 individually injected embryos per group (in 2 experiments). (**i**) Whole-embryo view of liposome distribution in kdrl:GFP transgenic embryos, 1 h after injection with DOPG and DSPC liposomes. Liposome accumulation for both formulations was observed in the primitive head sinus (PHS), common cardinal vein (CCV), posterior cardinal vein (PCV), and caudal vein (CV). (**j**) Tissue-level view of liposome distribution in kdrl:GFP transgenic embryos, 1 h after injection with DOPG and DSPC liposomes at 102 hpf. Liposome accumulation was observed in the entire caudal vein (CV), but only on the dorsal side of the PCV (dPCV, arrows). Reproduced with permission from [156], American Chemical Society, 2018.

**Figure 6 pharmaceutics-13-01615-f006:**
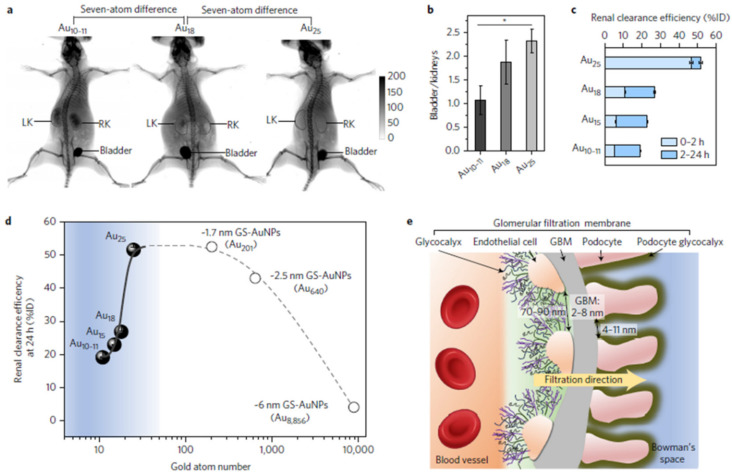
Renal clearance of different-sized AuNCs and schematic diagram of the glomerular filtration membrane. (**a**) whole-body X-ray images of mice after being intravenously (i.v.) injected with Au_10-11_, Au_18_, or Au_25_ at 40 min p.i. Although all three different AuNCs were cleared through the kidneys into the bladder, the smallest, Au_10-11_, shows much longer kidney retention than Au18, which in turn shows a longer kidney retention than Au_25_, even though there is only a seven-atom difference among these three AuNCs. LK, left kidney; RK, right kidney. (**b**) X-ray intensity bladder-to-kidney ratios of Au_10-11_, Au_18_, and Au_25_ at 40 min p.i., clearly showing that more Au_10-11_ and Au_18_ were retained in the kidneys than Au_25_. * *p* < 0.05, based on one-way ANOVA (*n* = 3 for Au_10–11_ and Au_25_; *n* = 4 for Au_18_). (**c**) Renal clearance efficiency of Au_10–11_, Au_15_, Au_18_, and Au_25_ at 0–2 h and 2–24 h after i.v. injection (*n* = 3 for Au_10–11_, Au_15_ and Au_25_; *n* = 6 for Au_18_). (**d**) Renal clearance efficiencies of Au_10–11_, Au_15_, Au_18_ and Au_25_, 1.7 nm (Au_201_), 2.5 nm (Au_640_), and 6 nm (Au_8856_) GS-AuNPs at 24 h p.i. versus number of gold atoms. Below Au_25_, the renal clearance efficiency decreased exponentially with the decreasing number of gold atoms in the NPs. (**e**) The glomerulus, an important component of renal filtration, is composed of kidney blood vessels, the glomerular filtration membrane, and Bowman’s space. The glomerular filtration membrane is composed of multiple layers: endothelial glycocalyx, endothelial cell, glomerular basement membrane (GBM), and podocyte. Podocytes are covered by a 200-nm glycocalyx. Generally, the fenestration between endothelial cells is 70–90 nm, the GBM junction is 2–8 nm, and the podocyte slits are in the range of 4–11 nm. With the combination of these layers, the size threshold for kidney filtration is ~6 nm. NPs or proteins with a hydrodynamic diameter (HD) of <6 nm can pass through the glomerular filtration membrane readily, but it is difficult for large particles to cross through. Reproduced with permission from [189] Nature, 2017.

**Figure 7 pharmaceutics-13-01615-f007:**
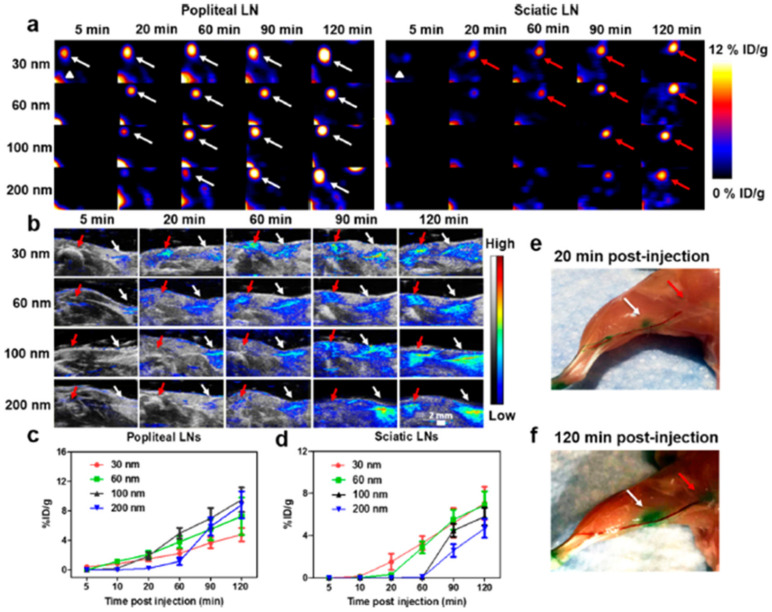
Representative (**a**) PET images and (**b**) overlaid coronal PA and ultrasound (US) images that illustrate size-dependent uptake in popliteal lymph nodes (LNs) and sciatic LNs at different time points post-injection. Quantitative analysis of the total PDI NP PET signal with uptake of different-sized NPs in (**c**) popliteal LNs and (**d**) sciatic LNs. The LN mapping is visualized after the footpad injection of PDI NPs at (**e**) 20 min and (**f**) 120 min post-injection. All white arrows in figures represent popliteal LNs, red arrows represent sciatic LNs, and white arrowheads represent injection sites. Reproduced with permission from [249], American Chemical Society, 2017.

**Figure 8 pharmaceutics-13-01615-f008:**
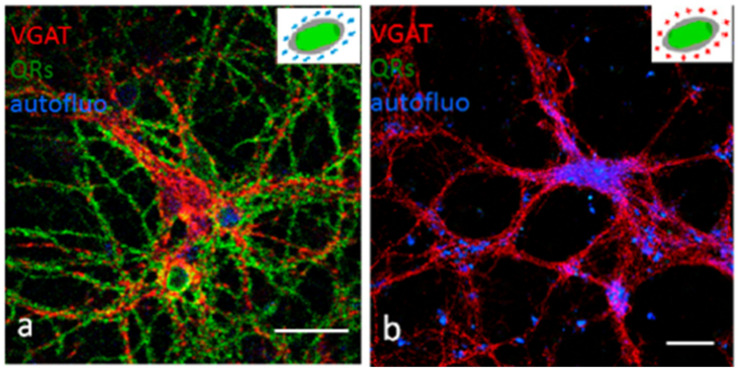
The effect of NP surface charge on their interaction with neurons: confocal microscope images of primary hippocampal neural cells incubated with 1 nM of negatively charged fluorescent QRs after 10 min of incubation at RT. Neurites and dendrites are covered by negatively charged QRs (**A**, green signal). Yellow represents the combination of NPs (green signal) and a neuronal marker (VGAT, red signal) and highlights the healthy condition of the entire neural network and the colocalization between QRs and synapses (MCC = 0.45 ± 0.06). (**B**) The same neuronal culture incubated with QRs that were identical in shape and size, but with a positive zeta potential; note the absence of QR fluorescence. These results are independent of QR size. Quantum rods (QRs), Manders’ correlation coefficient (MCC). Reproduced with permission from [276], American Chemical Society, 2017.

**Figure 9 pharmaceutics-13-01615-f009:**
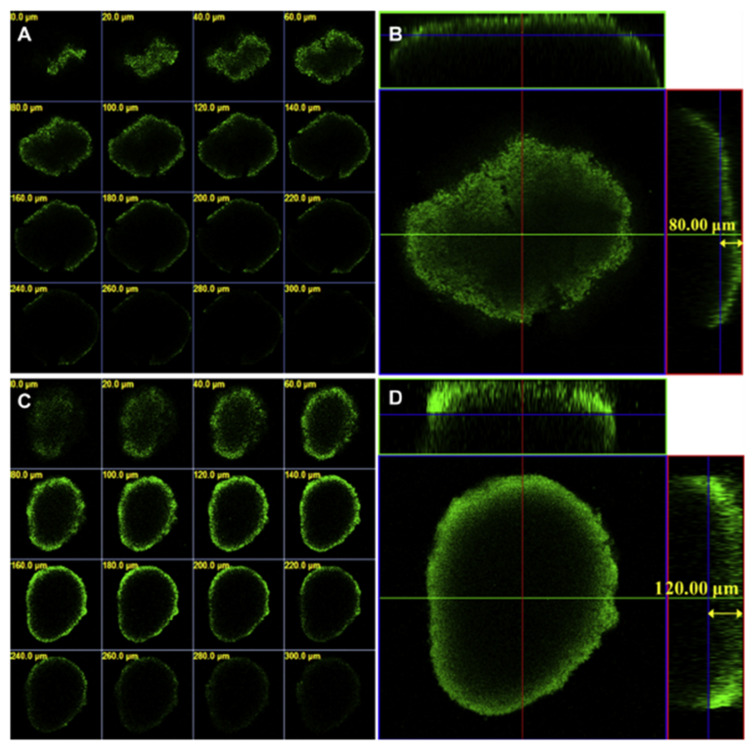
Penetration of coumarin-6-labeled NsP and Pep-NPs in 3D glioma spheroids. Z-stack images were obtained, starting at the top of the spheroids at 20-mm intervals, depicting the penetration of NPs (**A**) and Pep-NPs (**C**) for a total of 300 mm into the spheroids. Quantitative analysis of the penetration depth of NPs (**B**) and Pep-NPs (**D**). Pep-NPs penetrated much deeper, with a distance of 120.00 μm in tumor spheroids, and the fluorescence was distributed more extensively, whereas the fluorescence was mainly located at the edge of the spheroid for unmodified NPs, and the penetration depth was 80.00 μm. Reproduced with permission from [297], Elsevier, 2014.

**Figure 10 pharmaceutics-13-01615-f010:**
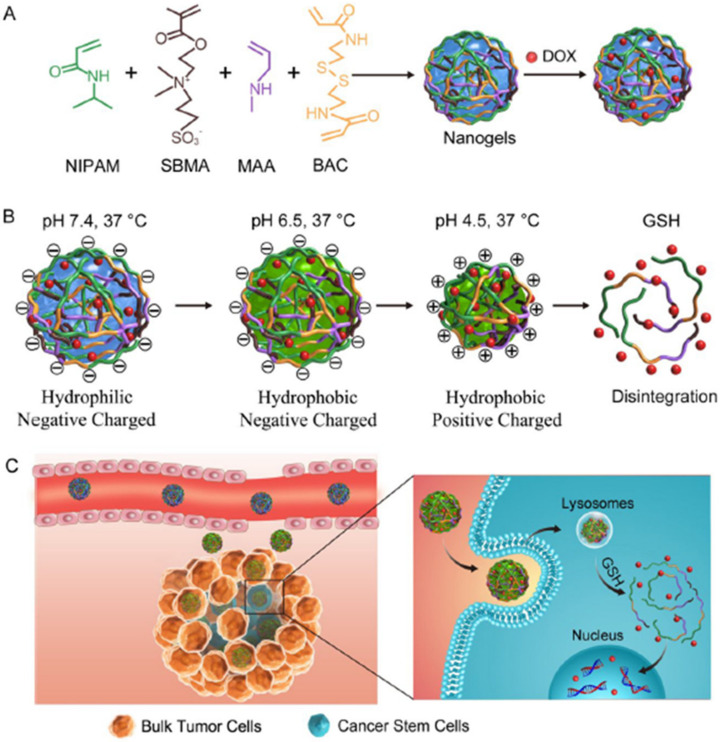
Schematic illustration of hydrophobicity-adaptive nanogels for programmed anticancer drug delivery. (**A**) Construction of the nanogels. (**B**) Characterization of the nanogels in response to the tumor microenvironment. (**C**) Schematic illustration of the in vivo transport process of the nanogels during anticancer drug delivery. Reproduced with permission from [317], Elsevier, 2018.

**Figure 11 pharmaceutics-13-01615-f011:**
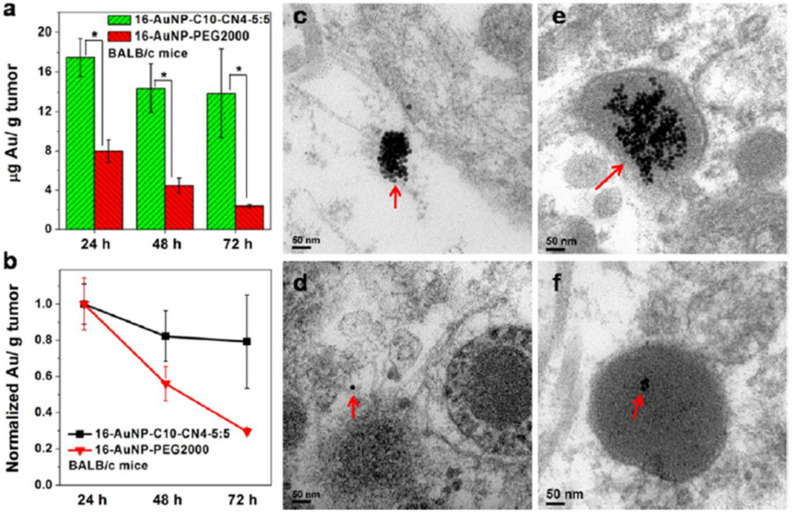
(**a**) Accumulation of 16-AuNP-C10-CN4-5:5 and 16-AuNP-PEG2000 in KB tumors in BALB/c nude mice at 24, 48, and 72 h post-injection. (**b**) Tumor uptake normalized at 48 and 72 h post-injection to that at 24 h (error bars represent mean (SD (*n* = 3); asterisk indicates significant difference, * *p* < 0.05). (**c**–**f**) Representative TEM images of sections of KB tumor tissue after injection with AuNPs for 24 h: 16-AuNP-C10-CN4-5:5 located in the interstitium (**c**) and lysosome of tumor cells (**e**) and 16-AuNP-PEG2000 located in the interstitium (**d**) and lysosome of tumor cells (**f**); red arrows indicate the AuNPs. PH-responsive aggregation was observed in acidic tumor spaces (**c**), and AuNPs appeared to be internalized effectively in tumor cells as aggregates in a similar manner to particles internalized in KB cells in vitro (**e**). Furthermore, two types of AuNPs exhibited different aggregation behaviors (**d**), and lysosomes of tumor cells revealed the presence of only a few small aggregates of 16-AuNP-PEG2000 (**f**). Each mouse was injected with 100μg of AuNPs. Reproduced with permission from [332], American Chemical Society, 2013.

**Table 1 pharmaceutics-13-01615-t001:** Pharmacokinetics studies of NPs. Abbreviations: polyelectrolyte nanocomplexes (NCs), cell-penetrating peptide (CPP), coating PLGA NPs with cationic octa-arginine (R8) peptide and specific anionic phosphoserine (Pho) (P-R8-Pho NPs). Human cervicovaginal mucus (CVM), luminescent porous silicon nanoparticles (LPSiNPs), superconducting quantum interference device (SQUID), black phosphorus Qds (BPQds), gadolinium-encapsulated graphene carbon nanoparticles (Gd@GCNPs).

Process	NPs	Application	Remarks	Ref.
Absorption	NCs	Insulin delivery	NPs possessed high epithelial absorption mediated by CPP and excellent permeation in mucus. The absorption of NPs on mucus-secreting epithelium cells was 20-fold greater than free insulin and improved the concentration of insulin in diabetic rats.	[108]
	Au	Cardiovascular disease	Inhaled AuNPs translocated from the lung to the circulation and selectively accumulated at vascular inflammation sites in human and animal models.	[109]
	P-R8-Pho	Insulin delivery	P-R8-Pho NPs improved absorption in the intestine in vivo and oral administration of insulin-loaded P-R8-Pho NPs exhibited 1.9-fold higher oral bioavailability, as well as better hypoglycemic effects on diabetic rats.	[110]
Distribution	Au	Cancer immunotherapy	Cellular distribution of 50-nm PEG coated AuNPs in the spleen, exhibiting the presence of NPs in B cells, T cells, granulocytes, and dendritic cells.	[111]
	PLGA	Drug delivery	PLGA-PEG blends caused quick diffusion in human CVM and increased vaginal distribution in mice in vivo.	[112]
	Qds	Biomedicine	In vivo biodistribution of indium-based water-soluble QDs in rats was measured up to 90 days and indicated the accumulation of QDs largely in the liver and spleen.	[113]
Metabolism	LPSi	Drug delivery	LPSi NPs were degraded in vivo over a period of 4 weeks and NIR imaging confirmed the elimination of NPs into the bladder at 1 h post-injection.	[114]
	Iron oxide	MRI	Magnetic nanocrystals were metabolized via the loss of their peculiar superparamagnetic properties using ferromagnetic resonance and SQUID measurements.	[115]
	PLGA	Drug delivery	Quantitative measurement of 200 nm and 500 nm PLGA degradation in liver and spleen after intravenous administration in mice.	[116]
	C3	Photothermal/photodynamic therapies	Lower toxicity and faster metabolic rate than Au nanorods.	[117]
Elimination	BPQds	Photodynamic therapy for cancer	BPQDs revealed proper stability, with no observable toxicity after PEG conjugation. They also eliminated rapidly through renal clearance due to their small size (5.4 nm) and showed great antitumor efficiency.	[118]
	Au	X-ray imaging of kidney dysfunction	X-ray imaging of renal-clearable AuNPs in nephropathic and normal kidneys. The ureteral obstruction inhibited the clearance of NPs via the ureter and slowed down the transportation of NPs from the medulla to the pelvis with improved cellular uptake.	[119]
	Cu_2_ZnSnS_4_	Tumor photothermal therapy	Peroxidase-like catalytic activity of nanocrystals could lead to oxidative lesions and kill cancerous cells. The rapid in vivo clearance of nanocrystals was confirmed via MRI, PA, and ICP-MS.	[120]
	Polycation	Metastatic breast cancer	Fluorination slowed the rate of renal clearance of NPs and helped to improve accumulations in tumor sites.	[121]
	Gd@GCNPs	Photodynamic therapy	Compared with C-dots, Gd@GC NPs indicated better T_1_ relaxivity, strong red fluorescence, high crystallinity, and suitable Gd encapsulation. These 50nm Gd@GC NPs accumulated in tumors via the EPR effect and were efficiently removed through renal clearance, resulting in no long-term toxicity.	[122]

**Table 2 pharmaceutics-13-01615-t002:** Physiochemical properties of NPs that determine toxicity.

Highlighted Factor	NPs	NP Features	Application	Remarks	Ref.
Size	Silica	20–80 nm	BBB	40-nm DOX-MSNPs showed higher permeability across the BBB, inhibited the undesired toxic side effects to normal brain tissue, and exhibited good anti-glioblastoma efficacy.	[199]
	PLGA	50 and 150 nm	Psoriasis	NPs of 50 nm enhanced sustained release, accumulation, and penetration, with deeper penetration into the psoriatic skin.	[200]
	Gold	1.5, 4, and 14 nm	Stem cell	AuNP of 1.5 nm was highly toxic, and AuNPs had size-dependent impacts on the pluripotency, viability, and neuronal differentiation potentials of stem cells.	[201]
	Nanodrug	10–90 nm	Cancer theranostics	A size-reducible self-assembled nanodrug significantly increased photodynamic effects, and deep tumor penetration was observed for smaller particles.	[202]
	Micelle	40, 90, 130, and 180 nm	siRNA delivery	The highest tumor retention and the best balance of cellular uptake and prolonged circulation were observed for 90 nm.	[203]
	PEG-PLGA	50, 100, and 150 nm	Chemoradiotherapy	Smaller NPs (sub 50 nm) penetrated the tumor more homogeneously, whereas they induced greater toxicity than larger particles.	[204]
	Gold	3.8 and 22.1 nm	Multidrug resistance cancer	TAT-modified AuNPs improved anticancer activity, and the larger NP was desirable to overcome multidrug resistance.	[205]
Shape	Carbon	SWCNTs and fullerenes	Angiogenesis	SWCNTs enhanced endothelial tubulogenesis, triggered integrin clustering, and activated focal adhesion PI3K signaling in endothelial cells, whereas the spherical NPs of fullerenols or fullerenols-DOX exerted a highly different antiangiogenic activity in zebrafish and murine tumor angiogenesis.	[206]
	Qds	Sphere and rod shapes	Tumor penetration	Nanorods showed faster tumor penetration compared to spherical NPs. Flexible nanorods possessed better circulation times than rigid rods.	[207]
	Hydroxyapatite	Sphere and rod shapes	Stem cell	Although no important impact on cell proliferation or migration was observed for both nanorods and nanospheres, NPs of spherical shape highly improved the osteoblastic differentiation of rat mesenchymal stem cells compared to nanorods.	[208]
	Quinoline–malononitrile	Sphere and rod shapes	Tumor targeting	The sphere-shaped NPs showed great tumor-targeted bioimaging ability than their rod counterparts.	[209]
	Polymer	Sphere and non-sphere	Drug delivery	Spherical NPs displayed higher cellular uptake than non-spherical particles.	[210]
	Silica	Short rod and long rod	In vivo areas	Long rod MSNPs were trapped in the spleen, whereas short rod particles were distributed in the liver and showed faster clearance rate.	[211]
	Gold	Sphere and star shapes	siRNA delivery	Spheres of 50 nm and 40-nm star NPs had much greater uptake efficiency than 13-nm sphere particles.	[212]
Surface charge	Silica	Positive charge	Drug delivery	Highly charged moieties were removed rapidly from the liver to GIT, whereas less charged moieties remained sequestered within the liver.	[213]
	Cellulose	Positive and negative	Bioimaging and drug delivery	The cationic cellulose nanocrystal (CNC)-rhodamine B isothiocyanate (RBITC) was uptaken via human embryonic kidney 293 and *Spodoptera frugiperda* cells without affecting the cell membrane integrity. At physiological pH, no significant internalization was observed for negatively charged CNC-fluorescein isothiocyanate (FITC).	[214]
	Fe_3_O_4_	Positive and negative	Biomedical applications	The cellular uptake of negative poly(acrylic acid) (PAA-MNPs) was under 50%, as opposed to almost 100% for cationic PEI-MNPs.	[215]
	Silicon	Positive and negative	Drug delivery	Cationic amine modified hydrophilic silicon NPs induced higher ATP depletion and genotoxicity than negatively charged hydrophilic and hydrophobic silicon NPs.	[216]
	Lipid	Positive	Cancer therapy	Decreasing the PEG density and increasing the surface charge could notably enhance the toxic role of cationic lipid NPs in systemic platelet activation and aggregation in vivo. Pretreatment of the mice with heparin enhanced the circulation time, were very efficient in preventing toxicity, and increased the anti-cancer drug delivery.	[217]
	Phospholipids	Positive, neutral, and negative	Ear drug delivery	Cationic PEG delivered DEX to inner ear hair cells with higher efficiency than Dex-sodium phosphate.	[218]
Functionality	Polystyrene	Amino groups	Leukemia	Amino-functionalized polystyrene NPs avoided proliferation in three leukemia cells and G_2_ cell cycle arrest.	[219]
	Polystyrene	Biotin	Drug delivery	Pretreated biotin-functionalized NPs showed 6- to 18-fold greater uptake and fusion protein control.	[220]
	Silver	Peptide	Antibacterial	The modified NPs indicated antibacterial activity against both Gram-negative (*P. aeruginosa*) and Gram-positive (*S. aureus*) bacteria, and eliminated antibiotic-resistant bacteria without toxicity to mammalian cells.	[221]
	CNT	Carboxyl groups	Biomacromolecules	Ideal photothermal encapsulation efficiency was observed for fibers constructed from 0.4 mg mL^−1^ of CNTs.	[222]
	Zirconium-metal–organic frameworks	DNA	Biosensor/bioimaging	DNA-functionalized NPs not only provided targeted imaging of cancer cells, facilitated the therapeutic delivery, and improved immunostimulatory effects, but also exhibited no toxicity.	[223]
	PLGA	Albumin	Tumor drug delivery	Only with a native conformation could albumin act as an ideal modifier for the tumor drug delivery of NPs.	[224]
	Coordination polymer	Bisphosphonates	Breast cancer	The modified NPs exhibited much greater affinity for bone in vivo and hydroxyapatite in vitro, decreased the osteocalastic bone destruction, and avoided the tumor growth effectively with notably reduced toxicity of cisplatin.	[225]
	Gadofullerene	Amino acid	Tumor vascular targeting	Excellent antitumor activity in hepatoma H22 models was reported for functionalized NPs. The use of amino acids enhanced the tumor inhibition rate up to 76.85%, and increased blood circulation time.	[226]
	MWCNT	Hydroxyl or carboxyl groups	Nanomedicine	Compared with pristine NPs, surface functionalization reduced the toxicity of MWCNTs and resulted in the intracellular uptake of particles.	[227]
	Silica	Amine, sulfonate, PEG, and PEI	Targeted drug delivery	In the presence of serum, sulfonate-functionalized dox-loaded MSNPs were effectively taken up into the cells and caused the highest cytotoxic and anti-proliferative effects in osteosarcoma.	[228]
Hydrophobicity	Amphiphilic	Hydrophobic	Immunoengineering	The hydrophobicity of amphiphilic poly(γ-glutamic acid)-graft-_L_-phenylalanine ethyl ester (γ-PGA-Phe) NPs influenced the uptake of encapsulated antigens by DCs. Compared to the conventional vaccine, the activation of DCs could be manipulated about 5- to 30-fold through adjusting the hydrophobicity of encapsulated antigens, and the effect on cellular immunity was reported to be almost 10- to 40-fold.	[229]
	Gold	Hydrophobic	Nanomedicine	AuNP surface hydrophobicity did not notably influence cellular uptake, whereas increasing NP surface hydrophobicity caused membrane damage and autophagy.	[230]
	Silver	Hydrophobic	Nanomedicine	The increase in hydrophobicity of Ag NPs in the presence of serum proteins resulted in higher cytotoxicity, cell uptake, and ROS production.	[231]
Aggregation	Maghemite		Cell therapy	The aggregation of NPs not only confounded global iron quantification and MRI cell detection but also displayed adverse impacts on chondrogenetic differentiation.	[232]
	Magnetic		Hyperthermia	The aggregation of magnetic NPs was independent of NP coating, cell type, and incubation time. One of the reasons for various magnetic behaviors in cell-induced aggregation may be attributed to the aggregation of magnetic NPs in a non-controlled way.	[233]
	Gold		Nanotechnology	Nanosized liposomes enhanced steric hindrance, which avoided the aggregation of AuNPs on a lipid membrane in a fluidic liquid-crystalline phase.	[234]

## Data Availability

Not applicable.

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
