# Peer review of "Safety Evaluation of Nanotechnology Products"

_pharmaceutics, 2021, doi:10.3390/pharmaceutics13101615_

Round 1

Reviewer 1 Report

The manuscript “Safety Evaluation of Nanotechnology Products” by Ghorbanali Sharifzadeh et al. showed recent advances in toxicity of nanotechnology products. They covered in vitro cytotoxicity assay and in vivo fate of nanomaterial. They also emphasized on environment, health and biohazards of nanomaterial.In my opinion it can be considered for publication with minor revision.

  1. In page no. 3, line no. 64 it should be joining of diagnostic devices.

  1. The author should cite reference for line 72-74 in page no. 4

  1. The author should cite reference for line 127-128 in page no. 6

  1. The author should cite reference for line 139-143 in page no. 6 and 7

  1. The author should cite reference for line 171-176 in page no. 8

  1. The author should cite reference for line 529, 3.2 (In vitro cell-based cytotoxicity assay) in page no. 25.

  1. The author should cite reference for line 666-667 in page no. 31.

Author Response

Reviewer 1

Author’s response

1. In page no. 3, line no. 64 it should be joining of diagnostic devices.

It has been corrected and shown in green colour: joining

2. The author should cite reference for line 72-74 in page no. 4

Line 72-74 were deleted.

There has been insufficient evaluation of nanomaterial interfaces for use with tissue and organs, even though several research projects have asked if nanoscale materials are to be treated as carcinogens or asbestos.

3. The author should cite reference for line 127-128 in page no. 6

Reference [29] is added to support this line.

[29] Duan J, Kodali VK, Gaffrey MJ, Guo J, Chu RK, Camp DG, et al. Quantitative Profiling of Protein S-Glutathionylation Reveals Redox-Dependent Regulation of Macrophage Function during Nanoparticle-Induced Oxidative Stress. ACS Nano 2016;10:524-538.

4. The author should cite reference for line 139-143 in page no. 6 and 7

Reference [33] is added and shown in green colour.

[33] Andón FT, Fadeel B. Programmed Cell Death: Molecular Mechanisms and Implications for Safety Assessment of Nanomaterials. Acc Chem Res 2013;46:733-742.

5. The author should cite reference for line 171-176 in page no. 8

Reference [41] is added and shown in green colour.

[41] Hou Z, Zhang Y, Deng K, Chen Y, Li X, Deng X, et al. UV-Emitting Upconversion-Based TiO2 Photosensitizing Nanoplatform: Near-Infrared Light Mediated in Vivo Photodynamic Therapy via Mitochondria-Involved Apoptosis Pathway. ACS Nano 2015;9:2584–2599.

6. The author should cite reference for line 529, 3.2 (In vitro cell-based cytotoxicity assay) in page no. 25.

Reference [72] is added.

[72] Magdolenova Z, Collins A, Kumar A, Dhawan A, Stone V, Dusinska M. Mechanisms of Genotoxicity: A Review of In Vitro and In Vivo Studies with Engineered Nanoparticles. Nanotoxicology 2013;8:233-278.

7. The author should cite reference for line 666-667 in page no. 31

Reference [5] is added.

[5] Xu G, Zeng S, Zhang B, Swihart MT, Yong KT, Prasad PN. New Generation Cadmium Free Quantum Dots for Biophotonics and Nanomedicine. Chem Rev 2016;116:12234−12327.

Reviewer 2 Report

This article entitled: Safety Evaluation of Nanotechnology Products reviews the safety implications of using nanotechnology products in a variety of applications. The increasing widespread use of nanomaterials has created the need to constantly keep under review the hazards associated with their deployment in a variety of settings. This review attempts to provide an overview of the safety considerations in the use of a range of nanotechnology products. However, it suffers from a number of deficiencies which need attention. The paper is certainly too long by a half and should be edited with the focus on the key aspects of safety and toxicological actions in relation to the use of the nanomaterials. The abstract should be re-written to reflect the areas covered in the review. Accurate definitions should be provided in the vocabulary section so that vital information is not missing. The focus of the review should be on the adverse effects of the products on biological systems and the environment, and less on the techniques that were used to obtain the results. Emphasis should be on the health and safety risks of exposure to these nanomaterials. Perhaps a narrow coverage of a selection of commonly used products may be more useful for the practitioner.          

Author Response

Reviewer 2

Author’s response

The paper is certainly too long by a half and should be edited with the focus on the key aspects of safety and toxicological actions in relation to the use of the nanomaterials.

Regarding the length of the paper, we agree with reviewer. However, deleting several contents of the paper can significantly affect its quality. We made a conscious effort to provide a review paper that could fully educate readers about the current safety challenges of nanomaterials. 

The abstract should be re-written to reflect the areas covered in the review.

Accurate definitions should be provided in the vocabulary section so that vital information is not missing.

These definitions are newly added to the vocabulary section and shown in green colour:

Apoptosis: alternatively known as Type-1 cell death, is a process of programmed cell death.

Necrosis: a non-programmed and unregulated cell death process.

Pharmacokinetics - demonstrates how organisms affect particular materials (e.g., nanostructures) after entering into the body via several processes (e.g., absorption, distribution, metabolism, and elimination).

Physicochemical properties - various physicochemical properties of NPs include size, shape, surface functionality, surface charge, composition, hydrophobicity, aggregation, and solubility.

Biohazards - infectious agents or hazardous biological materials that present a risk or potential risk to the health of people, animals, or the environment

The focus of the review should be on the adverse effects of the products on biological systems and the environment, and less on the techniques that were used to obtain the results.

The aim of this paper is certainly on the safety issue of nanomaterials. We believe that the presented techniques could provide better insights into the current challenges.

Round 2

Reviewer 2 Report

The changes made to the re-submitted review are superficial and do not address my main comments which were: The paper is certainly too long by a half and should be edited with the focus on the key aspects of safety and toxicological actions in relation to the use of the nanomaterials. The abstract should be re-written to reflect the areas covered in the review. Accurate definitions should be provided in the vocabulary section so that vital information is not missing. The focus of the review should be on the adverse effects of the products on biological systems and the environment, and less on the techniques that were used to obtain the results. Emphasis should be on the health and safety risks of exposure to these nanomaterials. Perhaps a narrow coverage of a selection of commonly used products may be more useful for the practitioner.

A review is not meant to be a thesis or a technical report but a succinct presentation of recent findings in the field with an indication of where the authors see future developments. I would recommend that the abstract is re-written once the following changes are completed:

Delete line 48 from structures to line 79 to risks.

Delete line 81 to 83 to asbestos.

Re-write lines 115-117 to make clear what you are trying to say.

Delete lines 145-152, these terms should be familiar to the readers of this journal.

Delete lines 154- 161 to signalling

Delete lines 167-173 to apoptosis.

Delete lines 187-193.

Delete lines 227-236.

Line 379, the title should not be of the technique but the type of information provided when it is deployed for the study of the distribution of the elements in the nanomaterial.

Delete 380-384 to 1nm.

Link sections 3.1.1 and 3.1.2 because they provide similar information.

Delete lines 504-507 to For example.

Delete lines 524-528 to drug delivery.

Re-write lines 539-542 to make clear what you are trying to say.

Delete lines 558 from Typically, through to line 571.

Delete line 597 from There are to line 606.

Delete lines 614-631.

Delete lines 632-666.

From line 667-1475, much of the information can be found in standard textbooks or recent review articles (see below references). The focus of the sections covered here is the presentation of recent information.

References

Ranjan, S., Dasgupta, N., Singh, S. et al. Toxicity and regulations of food nanomaterials. Environ Chem Lett 17, 929–944 (2019). https://doi.org/10.1007/s10311-018-00851-z

Ahmad, F., Wang, X., Li, W., Toxico-Metabolomics of Engineered Nanomaterials: Progress and Challenges. Adv. Funct. Mater. 2019, 29, 1904268. https://doi.org/10.1002/adfm.201904268

Daniel Mihai Teleanu , Cristina Chircov, Alexandru Mihai Grumezescu and Raluca Ioana Teleanu  Nanomaterials, 2019, 9, 96; doi:10.3390/nano9010096

Rachel Foulkes, Ernest Man, Jasmine Thind, Suet Yeung, Abigail Joy and Clare Hoskins, The regulation of nanomaterials and nanomedicines for clinical application: current and future perspectives  Biomater. Sci., 2020, 8, 4653–4664

Round 3

Reviewer 2 Report

Most of my recommendations have been addressed but I am disappointed that the authors have not taken their time to read over the paper and address some of the obvious mistakes in grammar and spelling. It is not the job of the reviewer to do this. A review paper as long as this will only make an impact if the readers can easily follow the train of thought of the authors. Below is a list of my recommended changes to improve the quality of  expression.

Abstract

Line 12, should read: Nanomaterials are now being used in a wide variety of biomedical applications.

Line 13, should read: Medical and health related issues, however, have raised major concerns, in view of the potential risks of these materials against tissue, cell, and/or organs and these are still poorly understood.

Line 22, should read: The focus is on understanding the role that the physicochemical properties of the nanomaterials play in determining their toxicity.

Line 95, delete: The nanomaterials…… eliminate.

Line 97, should read: ….degrade and eliminated.

Line 116, replace mode with forms.

Line 117, should read: ….is in the reduced form.

Line 121, should read: In contrast, such reactions do not occur in the presence of all NPs such as cerium oxide (CeO2). CeO2 NPs do not cause the formation of ROS….

Line146-147, should read: ….SSG alteration evermore so……

Line 183, should read: ….. through the Bak and Bax pathways by activating……

Line 211, should read: …. that fails to imitate the complex in vivo physiological conditions.

Line 211, delete …. Thus, real-time data….

Line 225, the abbreviation for quantum dots is QD not Qd.

Line 233, should read: … has been reported through the use of FA-CDS.

Line 322, should read: The researchers…

Line 338, should read: … suppress and/or elicit no immune response.

Line 341, Do you mean structure not surfaces?

Line 367, should be leading not leaing.

Line 452, delete,  … a device which ……….

Line 554, delete, They visually monitor….. results

Line 695, delete, little research…. and replace with: there has been few studies that have evaluated……

Line 1031, replace optimal with optimum

Title to Table 2, should read: Physiochemical properties of NPs that determine tocity.

Line 1038, should read: Compared with healthy tissues, the EPR effect results in more than a 50-fold increase in the accumulation of NPs.

Line 1053, should read: Concerted efforts have been made in cancer research to determine the optimum size of NPs, and to improve tumour treatment and diagnosis.

Line 1060, should read: small sized NPs.

Line 1177, should read: Typically, neutral and negatively charged NPs possess.

Line 1179, should read: ….. both neutral and negatively charged NPs display….

Line 1182, should read: … and are transferred through the cell membrane.

Line 1188, the word is apparently not aparetly.

Line 1283, should read: In contrast to pristine NPs.

Line 1285, should read: … has been reported.

Line 1291, should read: …. and homing in on the glioma.

Line 1675, delete the sentence, As interest……

Lines 1676-1678, delete the sentence.

Line 1678, should read: Nanostructured systems…. can be used in the fabrication of …. novel bio-products for use in a variety of applications. However, the deployment of such materials can pose health hazards therefore there is the need to acquire knowledge about potential risks when they are used in nanotechnology, biotechnology, electronics and medicine. Future developments will very much depend on a better understanding of the health and safety risks associated with the use of these materials.

Lines 1683-1684, delete This review has….

Line 1685, should read: To keep up with this demand, it is…. to design appropriate educational courses and provide consultancy services on a variety of aspects including strategic decision making…. to support decision making.

Line 1688, should read: Governments ….. should make use of the opportunities in order to upskill their workforce.

Line 1692, should read: ….. cost saving and reduce risk by ….

Lines 1692-1710, re-write this passage so that it is clear what you are trying to say.
